# Rapid metabolic shifts occur during the transition between hunger and satiety in *Drosophila melanogaster*

Daniel Wilinski [1], Jasmine Winzeler[1,6], William Duren[2,6], Jenna L. Persons[1], Kristina J. Holme[3], Johan Mosquera[2], Morteza Khabiri[4], Jason M. Kinchen[5], Peter L. Freddolino [2,4], Alla Karnovsky[2] & Monica Dus[1]

Metabolites are active controllers of cellular physiology, but their role in complex behaviors is less clear. Here we report metabolic changes that occur during the transition between hunger and satiety in *Drosophila melanogaster*. To analyze these data in the context of fruit fly metabolic networks, we developed Flyscape, an open-access tool. We show that in response to eating, metabolic profiles change in quick, but distinct ways in the heads and bodies. Consumption of a high sugar diet dulls the metabolic and behavioral differences between the fasted and fed state, and reshapes the way nutrients are utilized upon eating. Specifically, we found that high dietary sugar increases TCA cycle activity, alters neurochemicals, and depletes 1-carbon metabolism and brain health metabolites N-acetyl-aspartate and kynurenine. Together, our work identifies the metabolic transitions that occur during hunger and satiation, and provides a platform to study the role of metabolites and diet in complex behavior.

[1] Department of Molecular, Cellular and Developmental Biology, The University of Michigan, College of Literature, Science, and the Arts, Ann Arbor, MI 48109, USA. [2] Department of Computational Medicine and Bioinformatics, The University of Michigan Medical School, Ann Arbor, MI 48109, USA. [3] Graduate Program in Molecular & Integrative Physiology, The University of Michigan Medical School, Ann Arbor, MI 48109, USA. [4] Department of Biological Chemistry, The University of Michigan Medical School, Ann Arbor, MI 48109, USA. [5] Metabolon, Inc, Morrisville, NC 27560, USA. [6]These authors contributed equally: Jasmine Winzeler, William Duren. [7]These authors jointly supervised this work: Alla Karnovsky, Monica Dus. Correspondence and requests for materials should be addressed to A.K. (email: akarnovs@umich.edu) or to M.D. (email: mdus@umich.edu)

J acob and Monod's work on the *lac* operon showed that metabolites can actively control cellular physiology. Yet, for most of the last century, our understanding of metabolism has been confined to its energetic function. Nutrients and their metabolic by-products have energetic value because they provide animals with fuel and biomass to support cellular functions. However, metabolites also have informational value: they function both as *messengers* by carrying data about the nutrient environment and as *transducers* by directly controlling gene expression, proteostasis, and signal transduction[1–3]. In the last decade, the shift in our understanding of metabolites from fuel and passive by-products, to dynamic entities that control cellular activities have highlighted the potential implications of metabolic regulation in biology. While the role of metabolic signaling and reprograming has been studied in the fields of development[4], immunology[5], and cancer[6], we know less about how these processes impact the brain, especially in the context of complex behaviors.

Here we began tackling this question by quantifying the changes in metabolite levels during the transition between hunger and satiety in *Drosophila melanogaster* fruit fly heads and bodies. While the neuroendocrine pathways involved in hunger and satiety have been studied, the exact metabolite changes that occur during the transition to satiation are unknown; thus, mapping them is the first step to begin studying the role of metabolic signaling in a complex behavior such as feeding. To ask how diet composition influences metabolite levels, we measured the metabolic profiles of fasted and refed fruit flies fed a high sugar diet for different days. As with many omics studies, understanding how metabolites fit into different cellular pathways and vary across conditions is a major challenge. To this end, we created Flyscape, an open-access application for Cytoscape that visualizes metabolomics data in the context of *D. melanogaster* metabolic networks and integrates them with other omics data, such as transcriptomics and proteomics.

Using a combination of behavioral, metabolomics, and transcriptional studies and by employing Flyscape, we show that fly heads and bodies have largely non-overlapping changes in metabolic profiles between the two feeding states (fasted and refed), and that compared to bodies, heads seem tuned to rapid changes in glucose availability at both the metabolite and transcriptional levels. Consumption of a 30% high sugar diet rapidly dulls differences in metabolic profiles between fasted and refed flies and reprograms the way nutrients are assigned to pathways. Together this work provides a starting point to study the role of metabolism in complex behavior by allowing researchers to exploit a genetically tractable organism in studies of specific diet-linked disorders.

## Results

**Metabolic transitions between hunger and satiety.** Studies over the last decade have highlighted the importance of metabolite levels in directing cellular physiology in eukaryotic organisms[1,7–11], including in *Drosophila*[12], but how metabolites influence behavior is unclear. To begin investigating this question we set out to identify the metabolic changes that occur during the transition between hunger and satiety in *Drosophila melanogaster* fruit flies. While many studies have looked at the feeding behaviors of flies fasted for longer (24–48 h) or shorter (0–5 h) times as a proxy for hunger and satiety, we designed a feeding paradigm to specifically capture the transition between these two internal states. To do this, we fasted male flies for 24 h so that they missed their evening and morning meals[13,14], then fed them a meal of either agar (fasted) or 400 mM D-glucose agar (refed) for 1 h on the next day (Supplementary Fig. 1a for a schematic of the manipulation). To make sure that a single

meal of 400 mM D-glucose was sufficient to switch the behavioral state of flies from hungry to sated, we measured the feeding behaviors of fasted and refed flies using the Fly-to-Liquid-Food Interaction counter (FLIC). The FLIC records the real-time feeding interactions of the fly proboscis with the food five times per second[13]. While the majority of fasted flies (light green, > 90%) ate on the FLIC, only ~60% of refed flies interacted with the food (Fig. 1a). This was consistent with changes in their motivation to forage for food (Supplementary Fig. 1b, refer to control diet, CD). Over the 1-h period, fasted flies had over three times more feeding interactions than refed flies (Fig. 1b). The number of feeding events, defined as the continuous succession of five or more feeding interactions, was also higher compared to that of refed flies (Fig. 1c). Accordingly, the duration (sec) of each feeding event was longer in fasted flies (Fig. 1d); however, there was no difference in the interval between feeding events (min) and the time it took flies to initiate the first meal on the FLIC (Fig. 1e–f). Thus, our feeding paradigm alters the internal state of flies from hungry to sated in 1 h.

Previous work has broadly shown that the energy state of fasted flies is different compared to non-fasted flies[15]. Fasted flies have lower glycogen levels and hemolymph glycemia compared to non-fasted flies[16], and circulating glucose levels increase rapidly with refeeding[17,18], even if triglycerides do not change (Supplementary Fig. 1c, refer to CD). To identify the metabolites that change during the transition between hunger and satiety, we collected the heads and bodies of fasted and refed flies and performed Ultrahigh Performance Liquid Chromatography-Tandem Mass Spectroscopy (UPLC-MS/MS) across four different platforms. To ensure that metabolism was rapidly quenched during collection and to maximize the number of metabolites measured[12], we chose heads and bodies instead of individual tissues. Together, we measured 391 metabolites across 12 conditions (~60 samples, Supplementary Data 1). The median Relative Standard Deviation (RSD) was 3% and 8% for internal standards and biochemicals, respectively, while metabolites in each sample condition have a median value of 30% (Supplementary Data 1). To further assess the technical variation, we performed a Principal Component Analysis (PCA) on all of the data, including samples that were obtained by pooling both the head and bodies (Supplementary Fig. 2). Notably, the pooled samples clustered into a group that was intermediate between the heads and body samples, with a smaller intra-group spread than experimental groups.

**Flyscape: a tool to visualize *D. melanogaster* omics data.** As with most omics studies, the analysis of metabolomics data requires computational tools to interpret experimentally determined changes and place them into biological context. A common approach to achieving this involves mapping metabolites onto individual metabolic pathways[19,20]. However, since one metabolite can be incorporated into several pathways, it can be difficult to understand the impact of changes across many pathways. Though useful, this approach can lead to oversimplification. Our previously published tool Metscape[21] took a different approach. Metscape uses pathway information to build metabolic networks, thus allowing a more comprehensive view of the data. However, Metscape is human-centric and to the best of our knowledge none of the other existing tools support the analyses of *Drosophila melanogaster* metabolism. To overcome this limitation, we developed the user-friendly, open source tool Flyscape. Based on the Metscape frame work, Flyscape can simultaneously visualize *D. melanogaster* metabolomic and transcriptomic data (Supplementary Fig. 3a). Flyscape is a plug-in for the network analysis and visualization tool Cytoscape[22]. Flyscape uses the publicly accessible BioCyc[23] *D. melanogaster*

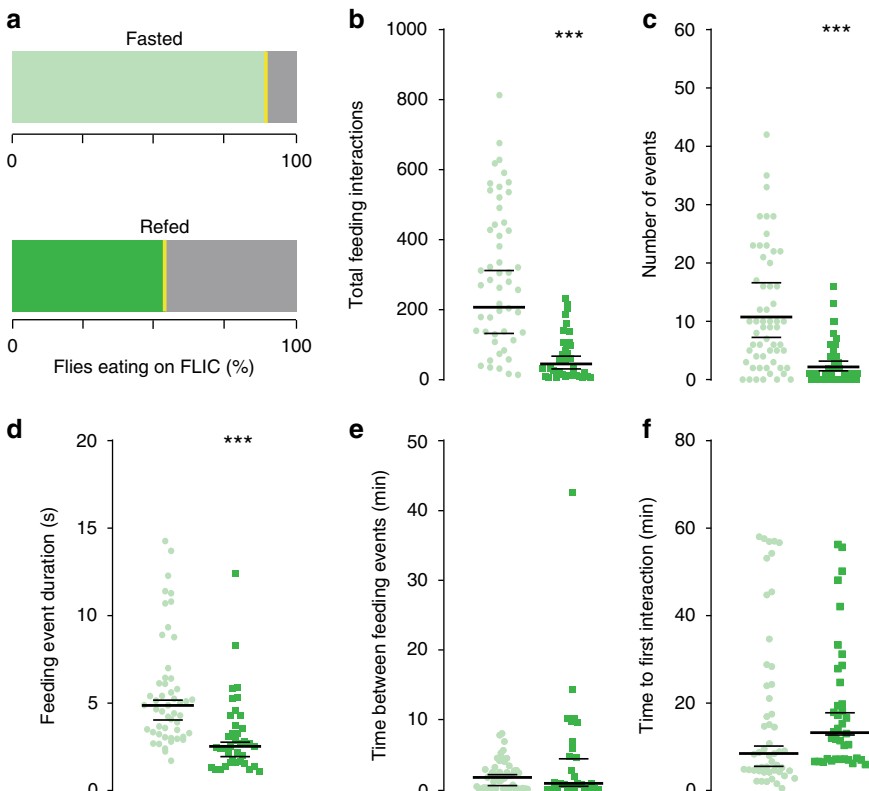

**Fig. 1** The feeding behaviors of fasted and refed flies. The feeding behavior of 24 h-fasted (light green, circles) and previously refed (darker green, squares) flies during ad-lib access to 5% sucrose on the FLIC for 1 h (See Supplementary Fig. 1 for a schematic). Individual data points are plotted in all cases. For panels **b**, **c**, which were analyzed using a Bayesian model (see Methods for details), bars show the average fitted value and the extent of a 95% credible interval; three asterisks indicate that the posterior probability from Bayesian analysis of a difference in the observed direction is greater than 0.999. For panels **d**–**f** plotted bars show a bootstrap-based median and extent of a 95% confidence interval; ***$p < 0.001$ using a stratified bootstrap. Fasted $n = 48$, refed $n = 38$ biologically independent animals. Source data are provided as a Source Data file. See Methods for a detailed description of the behavioral quantification and analysis. **a** The percentage of fasted and refed flies that interacted with food on the FLIC (green shades), compared to those that did not eat (gray shades). **b**–**f** The feeding behavior of flies during the 1-h ad-lib access to 5% sucrose quantified as **b** the total number of feeding interactions; **c** the number of feeding events (defined as five of more consecutive feeding interactions above threshold); **d** the mean duration (sec) among feeding events; **e** the mean time between 2 or more feeding events (min); **f** the latency to interact with food (min), calculated as the first feeding interaction initiated on the FLIC. For panels **d**–**f**, replicates with no feeding events were excluded (for panels **b**, **c** the no-interaction events are explicitly accounted for in our zero-inflated negative binomial model). Source data are provided as a Source Data file

metabolic pathway database with gene annotations collected from Flybase[24]. The Flyscape database contains 4856 metabolites, 2326 enzymes, 15,577 genes, and 2827 reactions. Flyscape can generate four types of network graphs with varying complexity: compound networks where compounds are represented as nodes and reactions are represented as edges (Supplementary Fig. 3b), compound-reaction (Supplementary Fig. 3c), compound-gene (Supplementary Fig. 3d), and compound-reaction-enzyme-gene networks (Supplementary Fig. 3e). In the last three types of networks the respective entities are represented as nodes. Notably, in all types of networks each node is unique and nodes from different pathways can be connected. Further, the networks can be refined by creating subnetworks based on pathways or compounds of interest. This feature allows visualization of changes across different pathways and conditions in a simple, intuitive way. In addition, Flyscape integrates metabolomics data with other types of omics data to build a network of the physiological landscape of the tissue or cell under different conditions.

**Bodies and heads show unique metabolic shifts**. We analyzed the metabolic profiles of the bodies and heads of refed and fasted flies. PCA showed that the variance between these two datasets was largely due to the feeding state (Supplementary Fig. 4a, b). In

the bodies, 61 metabolites changed in abundance between refed and fasted flies (Fig. 2a). Out of these 31 metabolites were increased and 30 metabolites decreased (Welch's $t$-test, FDR < 0.1) in refed flies. Compounds higher in refed flies (Fig. 2a) reflect an increase in glucose availability, catabolism (acetyl-CoA, malate, fumarate, lactate), utilization for post-translational modifications (UDP-N-acetylglucosamine, N6-carboxymethyllysine), and storage into glycogen (maltose). Conversely, fasted flies showed an increase in glutamate, α-ketoglutarate, tyrosine, and medium chain fatty acids, indicating higher catabolism of fatty acids, branched-chain amino acids, and ketone bodies for energy (Fig. 2a). Fasted flies also had changes in purine and pyrimidine catabolism, the urea cycle, and arginine and proline metabolism. We next used Flyscape to visualize changes in the tricarboxylic acid cycle (TCA cycle KEGG: dme00020) between refed and fasted fly bodies (Fig. 2b). The size of the compound nodes (hexagons) reflects changes in metabolite abundance between refed and fasted flies, (salmon-colored hexagons represent compounds that were not measured), and the green outlines highlight statistically significant measurements. While only acetyl-CoA, malate, and fumarate increased significantly, this visualization shows that nearly all compounds in the TCA cycle were higher in refed compared to fasted flies,

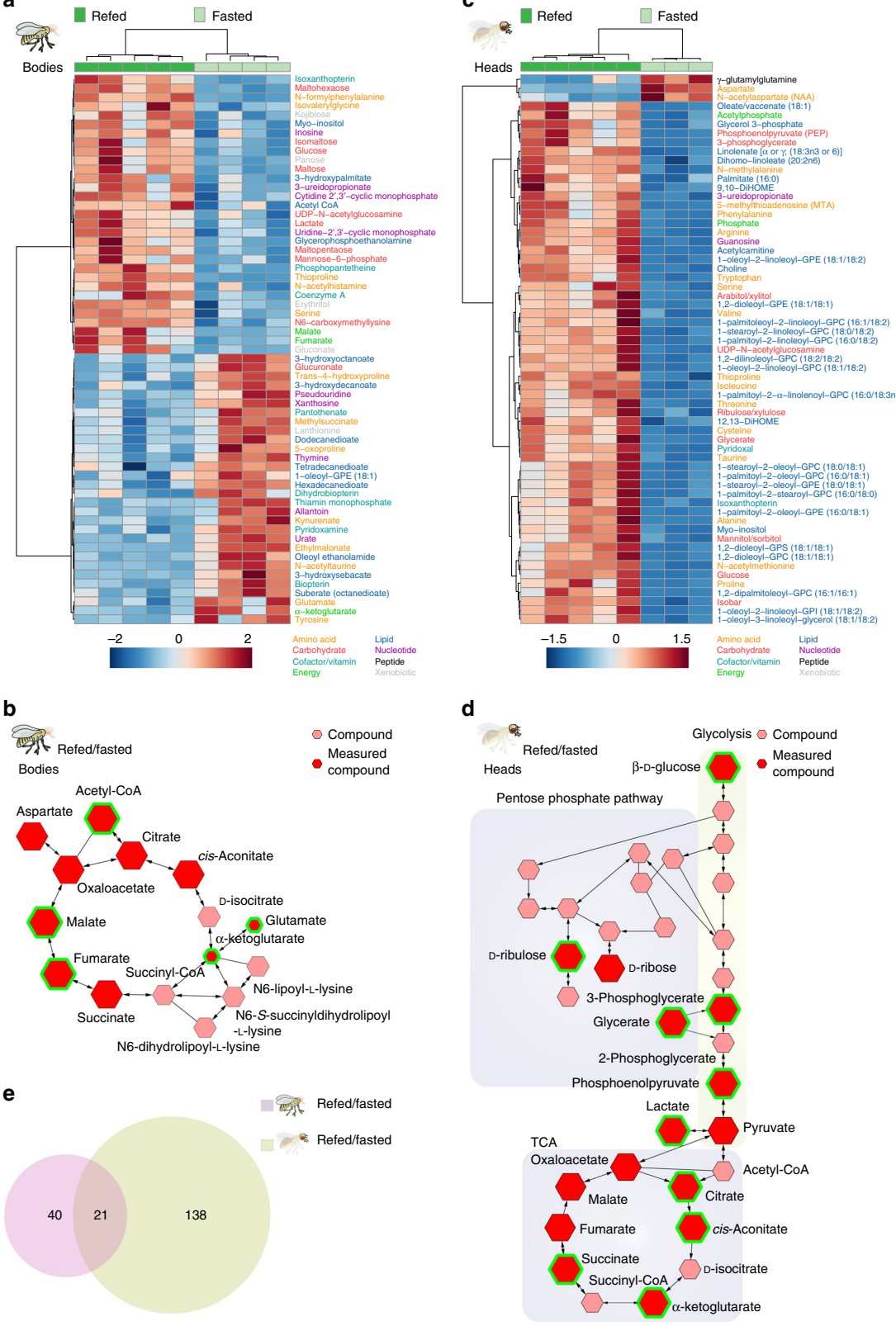

while, α-ketoglutarate and glutamate were higher in fasted flies. Thus, Flyscape offers a contextualized visualization of the changes in cellular energetics between the two feeding states (Fig. 2b) compared to the differential displays of Fig. 2a. In our data, Flyscape reveals a change in the TCA cycle that is easy to appreciate: it is fueled by glucose in refed flies and by glutamate through α-ketoglutarate in fasted flies. Taken together these data provide evidence of a shift from β-oxidation/fatty acid metabolism and ketosis during the fasted state to increased glucose utilization after refeeding.

We next asked if the metabolic profiles found in bodies also occurred in heads when flies were refed. Fly heads contain a

**Fig. 2** The shifts in metabolic profiles in the bodies and heads of fasted and refed flies. **a** Heatmap of the 61 metabolites changed in the bodies of flies between the fasted and refed conditions; Welch's *t*-test, FDR < 0.1. Normalized metabolite levels were clustered by compound (rows) and biological replicate (columns). The names of metabolites are colored according to their metabolic classes (bottom). The heatmap indicates positive (red shades) and negative (blue shades) normalized compound levels. **b** A Flyscape network showing the changes in the TCA cycle in the bodies of refed and fasted flies. The size of the compound nodes (hexagons) reflects changes in metabolite abundance (up or down) between refed and fasted flies, salmon-colored hexagons represent compounds that were not measured, and the green outlines highlight statistically significant measurements (Welch's *t*-test, FDR < 0.1). **c** Heatmap of the top 61 (out of 159) metabolites that change in the heads of fasted and refed flies (Welch's *t*-test, FDR < 0.1). Normalized compound levels were clustered by compound (rows) and data replicate (columns). The names of metabolites are colored according to their metabolic classes (bottom). The heatmap indicates positive (red shades) and negative (blue shades) normalized compound levels. **d** Flyscape network showing the metabolites changing in glycolysis, pentose-phosphate pathway, and TCA cycle in refed vs. fasted fly heads. The size of the compound nodes (hexagons) reflects changes in metabolite abundance between refed and fasted flies, red and salmon-colored hexagons represent compounds that were and were not measured, respectively, and the green outlines highlight statistically significant measurements (Welch's *t*-test, FDR < 0.1). **e** Venn diagram showing the overlap in the metabolic shifts between the bodies of fasted and refed flies (lavender shade) and heads (green shade) of flies. Welch's *t*-test, FDR < 0.1. Source data are provided as a Source Data file

multitude of tissue types, such as muscle and fat, but the majority of the head capsule is occupied by the brain's ~150,000 neurons and glia. Of note, there was greater variation in the heads of refed compared to fasted flies, likely reflecting natural variation in food consumption between animals during the refeeding period (Supplementary Fig. 4b). Of the 391 compounds measured, 185 changed between the fasted and refed states in fly heads (Welch's *t*-test, FDR < 0.1). The largest classes of compounds changed were lipids (85), amino acids (43), and carbohydrates (19) (Fig. 2c and Supplementary Data 1). Among the fats, we measured a large number of ether lipids, such as glycerophosphocholine (GPC) and glycero-3-phosphoryl ethanolamine (GPE), which are overrepresented in the fly and human brains[25–27]. Three times more compounds changed in heads between feeding states compared to bodies (159 vs. 61) and only 21 compounds were in common between the two (Fig. 2e). In contrast to bodies, all but three metabolites decreased with fasting in heads (Fig. 2c). Like the bodies, the metabolic profiles of refed heads showed higher glucose availability and increases in glucose-related products. However, heads had high-fold increases in glycolytic intermediates (KEGG: dme000103, phosphoglycerate, phosphoenolpyruvate) and end-products (lactate and pyruvate), and trends and increases in pentose-phosphate metabolites (KEGG: dme00030) indicating a differential and likely increased use of glucose (Fig. 2d). Glycogen intermediates were also increased suggesting higher glucose demand in heads. Higher levels of TCA cycle metabolites (citrate, aconitate, and α-ketoglutarate) and the pyruvate-acetyl CoA intermediate acetylphosphate suggest that carbon utilization into the TCA cycle may occur primarily from pyruvate derived from carbohydrate sources in the refed state (Fig. 2d). Finally, lipid metabolism was also rapidly altered by refeeding with increases in fatty acids as a class (Fig. 2c).

We also observed differences in neurotransmitters and their biosynthetic intermediates that were specific to heads (Supplementary Fig. 5). Gamma-aminobutyrate (GABA), glutamate, choline (the precursor for acetylcholine), and N-acetylserotonin were elevated in the heads of refed flies. In contrast, aspartate and N-acetylaspartate (NAA) were higher in fasted fly heads. γ-glutamylglutamine was also higher in fasted fly heads (Fig. 2c). This, together with the elevated levels of glutathione (Supplementary Fig. 5), may reflect higher oxidative stress in fasted fly heads (compared to bodies, where no change was observed). In conclusion, while both heads and bodies showed an increase in energy availability with refeeding, their metabolic profiles were largely non-overlapping, with higher glucose metabolism changes in heads.

**Transcriptional changes in brains of sated and refed flies.** Since we observed rapid and high magnitude variations in glucose metabolism in heads compared to bodies (Fig. 2), we wondered whether these metabolic changes could be supported by an increase in nutrient availability in this tissue, consistent with known changes in glucose in the hemolymph upon refeeding[18,28]. To probe this question, we measured changes in RNA abundance from the central brains of fasted and refed flies 1 h after feeding to allow for transcriptional responses to occur. To better focus on fast vs. slow responses to satiation, we also collected brains from flies that consumed their regular morning meal[14] and were never fasted (here termed sated, see Methods). We identified 30 transcripts that changed in the central brains of refed/fasted flies (Fig. 3a, Supplementary Fig. 6a–c, Supplementary Data 2, FDR < 0.05 by Wald test) and 113 in sated/fasted flies (Fig. 3b, Supplementary Fig. 6a, d, e, Supplementary Data 2, FDR < 0.05 by Wald test). The RNA levels of sugar transporters and other Solute Carriers (SLC) homologues were higher in both refed and sated flies, while the transcript levels of the *SLC17* (*CG3036, Picot, CG6978*), and *SLC36* (*CG7888*) transporters higher in fasted brains. Fasted brains showed higher levels of lipases (*brummer, CG5966*) and branched-chain amino acid (*CG1673*) and purine metabolism (*Gart* and *CG11089*) (KEGG: dme00230). Refed brains had increases in enzymes for lipid (*Lsd-1*), neurotransmitter (*CG12116*), purine (*Prat2*), and folate metabolism (*CG8665*). Interestingly, only 11 transcripts overlapped between the refed and sated conditions (Fig. 3c, Supplementary Data 2), suggesting that while transcriptional responses to refeeding occur rapidly, they are also distinct from those of sated flies. With the exception of the *Hormone receptor-like 38* (*Hr38*), transcription factors (*Cabut*[29], *fruitless*, and *doublesex*) showed differential abundance only in sated flies. Overall these changes are consistent with the increase in catabolism of amino acids and nucleotides we observed in the metabolomics data and the higher availability of nutrients in refed flies. To see if any of the metabolite and RNA level changes were linked, we used Flyscape to integrate our head metabolomics and brain RNA-sequencing data by looking for significant genes and metabolites in the same network. This analysis connected the gene *CG1673* to the conversion of the branched-chain amino acids (BCAA, leucine, valine, and isoleucine) and α-ketoglutarate to branched-chain keto acids (BCKA) (Fig. 3d). *CG1673* RNA levels were higher in fasted brains and decreased with satiation, matching an increase in BCAA levels. Indeed, *CG1673* is annotated as a branched-chain amino acid transaminase in Flybase, but has not been experimentally linked to a metabolic pathway. Our analysis supports the annotation of this gene and shows how the multi-omic feature of Flyscape can be used to identify and narrow down candidate

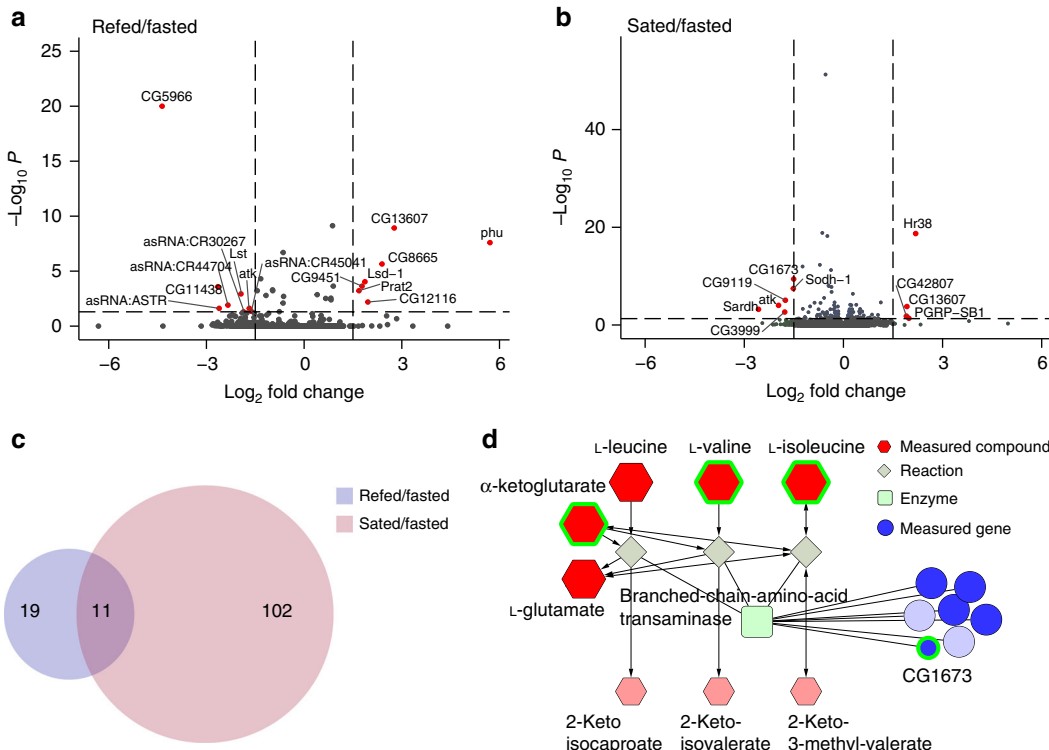

**Fig. 3** RNA abundance changes of genes involved in nutrient transport and metabolism with refeeding. **a, b** Volcano plots showing the significant changes (Wald test, FDR < 0.05) in transcript levels between **a** refed vs. fasted and **b** sated vs. fasted brains. See main text and Methods for details on the feeding manipulations. The horizontal dotted line defines the $p$-value cutoff of 0.05, and the vertical lines indicate a $\log_2$ fold change of ±1.5. Red circles indicate transcripts that pass $p$-value and $\log_2$ fold cutoffs. **c** Venn diagram showing the overlap in the transcripts that change between the refed/fasted (purple shade) and the sated/fasted (pink shade) conditions. **d** A partial Flyscape network made using both the metabolomics (Fig. 2) and the RNA-sequencing data (this figure) showing the metabolites and genes that change between the fasted and refed conditions in branched-chain amino acid metabolism. The size of the compound nodes (red hexagons) reflects changes in metabolite abundance (up or down) between refed and fasted flies, and salmon-colored hexagons represent compounds that were not measured. The size of the gene nodes (blue circles) represent the sign of changes in RNA abundance between fasted and refed flies, (light blue circles are genes that were not measured). Green squares represent the enzyme type. Compounds with an FDR < 0.1 by Welch's $t$-test and genes with a corrected $p$-value < 0.05 by Wald test are outlined in green. Source data are provided as a Source Data file

genes, especially enzymes, to test with functional experiments. This, together with the availability of genome wide tools to modulate gene expression in *Drosophila*, will help to map specific metabolic pathways and study their function.

**SD dulls metabolic transitions in fasted and refed flies.** Since metabolite levels change depending on diet composition, we investigated the effects of a high sugar diet on the metabolic and behavioral transition between hunger and satiety. The influence of a high sugar diet on obesity, metabolic syndrome, and insulin resistance has been widely studied in *Drosophila*[30,31]; we also recently found that consumption of this diet decreases the sensitivity of the sweet gustatory neurons to sugar, which alters feeding patterns and promotes diet-induced obesity[14].

We first examined the effect of a high sugar diet (SD, 30% sucrose) on the metabolome of the fasted and refed bodies compared to those of flies fed a control diet (CD, 5% sucrose). We observed 54/377 compounds changing in the refed state (CD/SD), and 149/381 compounds in the fasted state (CD/SD, Welch's $t$-test, FDR < 0.1) (Fig. 4a, c; Supplementary Fig. 7a, b and Supplementary Data 1). Compounds in the lipid and energy categories were elevated in both fasted and refed SD bodies (Fig. 4b, d), including long and medium chain fatty acids (acylcarnitines and triglycerides), glucose, lactate, and pyruvate. Consistent with this, flies on the SD had higher levels of triglycerides when these were measured using a colormetric assay

(Supplementary Fig. 1c). Several amino acids and compounds known to increase in the plasma of fasted humans with obesity also increased in flies on a SD, such as, glutamate, α-ketoglutarate, cysteine, and aspartate[32–34] (Supplementary Fig. 7). Metabolites in the hexosamine biosynthesis pathway were also elevated in both fasted and refed SD flies (Supplementary Fig. 7a, b). In contrast, the levels of acetyl-CoA and nucleotides were decreased on a SD compared to flies on a CD (Fig. 4b, d and Supplementary Fig. 7a, b). In accordance with this, we observed a decline in pentose-phosphate metabolites and an increase in ribose 5-phosphate, which may reflect changes in pentose-phosphate metabolism (KEGG: dme00030) with a high sugar diet (Supplementary Fig. 7a, b).

We next asked how a sugar diet influences the metabolic shift between hunger and satiety by comparing the metabolome of refed vs. fasted flies on a CD or SD. Principal component analysis of the bodies of flies on a SD revealed that two groups were well separated (Supplementary Fig. 4c). While biological replicates from each diet cluster together, the 95% confidence intervals overlap (lines), suggesting that a portion of the variance between the fasted and sated state is minimized when flies consume a high sugar diet. This is consistent with the finding that only 14 metabolites change between refed and fasted files on a SD (Fig. 4e), which is striking considering that 61 metabolites change in the bodies of refed and fasted flies fed a CD (Fig. 4e). To examine this phenomenon further, we considered the overall distribution

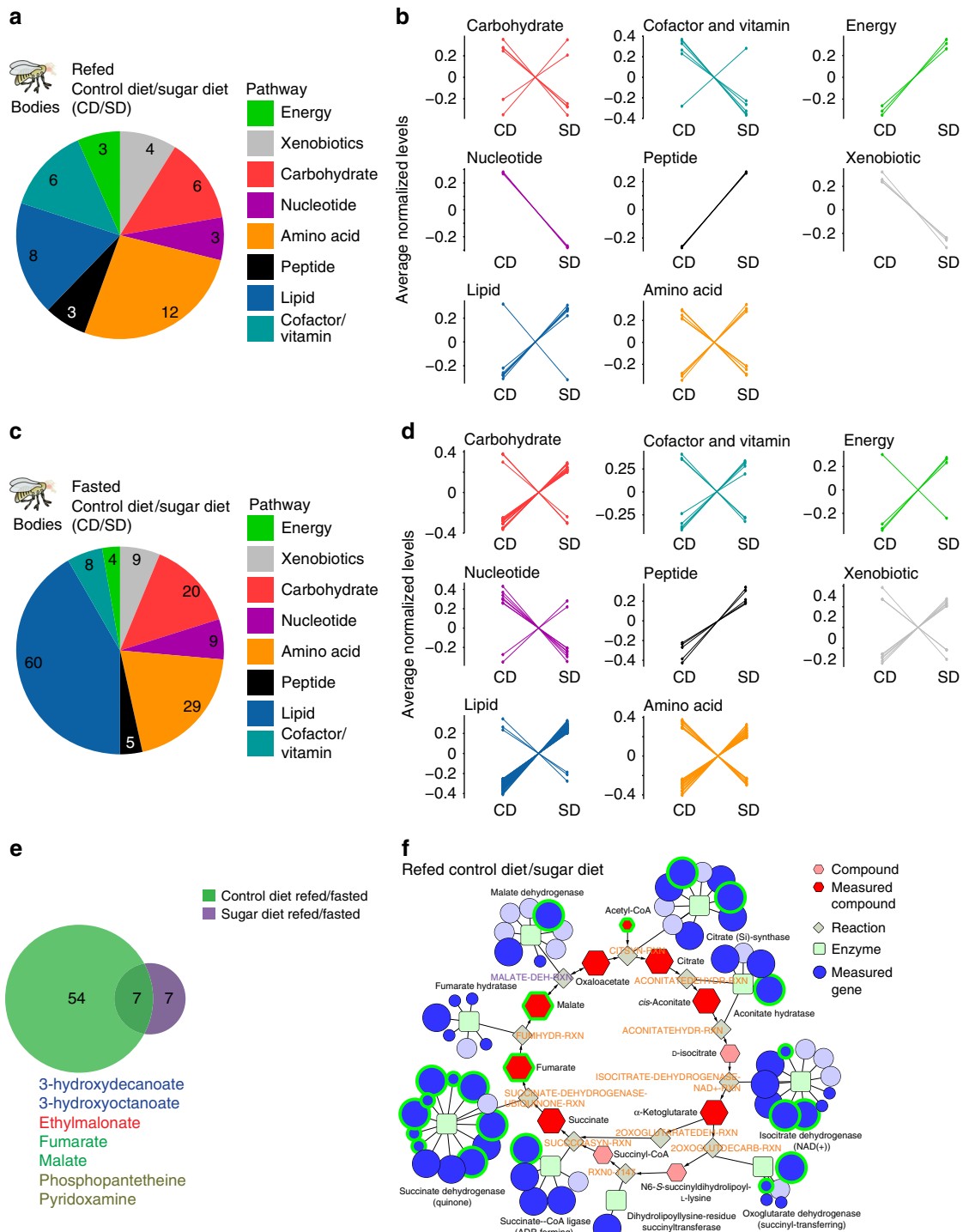

**Fig. 4** Consumption of a SD alters the metabolic profiles of fasted and refed flies. **a**, **b** The effect of a high sugar diet on the metabolite classes (labeled by different colors) in the bodies of **a**, **b** refed and **c**, **d** fasted flies. Control diet (CD) and high sugar diet (SD). Welch's *t*-test, FDR < 0.1. **b**, **d** The normalized levels of the compounds, grouped by class, that differ between **b** the refed CD and SD fly bodies (45 compounds), and **d** the fasted CD and SD fly bodies (144 compounds). Color scheme defines metabolite classes listed in **a**. **e** Venn diagram showing the overlap in the metabolic compounds that change between the refed and fasted conditions in the bodies of flies fed a CD (green shade) or SD (purple shade). Metabolites shared between flies on a CD and SD are listed and colored according to class as in color scheme from **a**. **f** A Flyscape network made by using compounds changed in bodies of refed flies fed a CD or SD (panels **a** and **b** in this figure) and RNA sequencing generated by another study (see main text and methods). The size of the compound nodes (red hexagons) reflects changes in metabolite abundance (up or down) between refed and fasted flies, and salmon-colored hexagons represent compounds that were not measured. The size of the gene nodes (blue circles) represent the magnitude of change in RNA abundance between fasted and refed flies, (light blue circles are genes that were no measured). Green squares represent the enzyme. Compounds with an FDR < 0.1 by Welch's *t*-test and genes with a corrected *p*-value < 0.05 by Wald test are outlined in green. Source data are provided as a Source Data file

of changes in abundance between the refed and fasted conditions (Supplementary Fig. 8a). The distribution is right shifted for CD, but centered at zero for SD, demonstrating that the changes in metabolite levels occurring during refeeding were essentially absent in the SD flies. Among the seven compounds that change independently of diet there are TCA cycle compounds, fumarate and malate and β-oxidation intermediates, 3-hydroxydecanoate and 3-hydroxyoctanoate (Fig. 4e). Refed SD bodies also showed a less robust increase in glycolysis upon refeeding, consistent with an increase in energy stores.

To better visualize the metabolic changes with the high sugar diet, we generated a Flyscape network comparing changes in TCA cycle metabolites in refed flies on a CD or SD (Fig. 4f) and plotted RNA-sequencing data generated from a previous study[35] (Supplementary Fig. 9a–c, Supplementary Data 2; see Methods for differences in the diets). Flies on a SD showed increases in fumarate, malate, oxaloacetate glutamate, and α-ketoglutarate upon refeeding, but lower acetyl-CoA levels, suggesting a possible change in TCA cycle function in refed flies on a SD. Flyscape also identified a number of enzymes involved in these pathways whose RNA abundance was changed by diet[35].

**The effects of SD on feeding behaviors**. Since fasted and refed flies on a SD have different body metabolic profiles compared to flies fed a CD, we examined its impact on acute feeding and foraging behaviors. To more carefully define the effects of diet exposure, we measured the feeding behaviors of age-matched male flies fed a CD or SD for 2, 5, and 7 days (SD2, SD5, and SD7). As in Fig. 1, we first fasted flies for 24 h, and then provided them with a meal of agar (fasted) or 400 mM D-glucose agar (refed) on the next day for 1 h (Supplementary Fig. 1a). We then assessed the effect of these manipulations on feeding behavior using the FLIC.

Fasted flies on a SD maintained their motivation to seek food (Supplementary Fig. 1b), which is consistent with the observation that diet had no effect on the percentage of refed or fasted flies eating on the FLIC (Fig. 5a). On both a CD and SD fasted flies had more feeding interactions and feeding events compared to refed flies (Fig. 5b, c), suggesting that the immediate responses to energy deprivation are maintained on a SD. However, fasted flies on a SD for 5 and 7 days interacted with their food less compared to fasted CD flies (Fig. 5b, SD2: $P = 0.92$, SD5: $P = 0.95$, 7D: $P = 0.96$ and Fig. 5c SD2: $P = 0.84$, SD5: $P = 0.96$, SD7: $P = 0.93$, posterior probability from Bayesian analysis), which is interesting, considering that SD flies have higher levels of energy and lipid metabolites and fewer metabolites changing between the fasted and refed states (14 vs. 61 metabolites, Fig. 4e). In addition, we find little difference in feeding event duration between fasted and refed (Fig. 5d). This suggests that the number of events drives the total feeding interaction, not event duration. Consequently, time between feeding events did not change dramatically between diets (Fig. 5e). However, a SD increased the time to initiate feeding (Fig. 5f). Thus, while flies on a high sugar diet still showed a clear behavioral response to fasting, this was smaller compared to that of CD flies, and the difference in feeding behaviors of hungry and sated flies was also decreased.

**SD alters metabolic responses to fasting and refeeding**. Since a sugar diet dulled the metabolic changes and blunted the magnitude of the behavioral transition between hunger and satiety states, we next investigated its effect on the metabolic profiles of heads. PCA showed that the variance between datasets was largely due to the feeding state, and the 95% confidence intervals are completely overlapping by SD7 (Supplementary Fig. 4d–f). First, we measured the number of metabolites changing between the

fasted and refed state. On a control diet, 159 metabolites differed between the refed and fasted conditions (Fig. 2e); however, at 2, 5, and 7 days on a SD, 10, 33, and 0 metabolites, respectively, changed in heads between fasted and refed flies (Fig. 6a and Supplementary Data 1). These metabolites did not belong to a single class in particular; instead the metabolic difference between fasted and refed heads collapsed rapidly with consumption of a SD (Fig. 6a, *blue bars*). As with the body data, the distribution of fold changes in metabolite levels between refed and fasted conditions in heads on a CD was strongly right shifted, but centered on zero for flies fed a SD (Supplementary Fig. 8b). Interestingly, the differences in neurochemical levels between fasted and refed flies also disappeared when flies ate a SD (Supplementary Data 1).

To examine how a high sugar diet alters the metabolite profile of fasted and refed state, rather than the transition between the two, we clustered compounds (refed 180 and sated 132, ANOVA FDR > 0.1) from heads that changed across CD, SD 2, 5, and 7 days (Supplementary Fig. 10 and Supplementary Fig. 11). We expected to find a pattern of smooth transitions where classes of metabolites gradually increased or decreased with longer exposure to diet. To our surprise, however, we found only a few gradual transitions (Supplementary Fig. 10a). In contrast, new metabolic profiles that characterize the fasted and refed states over different days on the SD emerged (Supplementary Fig. 10 and Supplementary Fig. 11). For example, less than half of the compound levels that define the refed state on a CD were maintained in the refed SD2 and SD5 flies (Supplementary Fig. 10b–d); instead, new refed metabolic profiles arose at 2 and 5 days (Supplementary Fig. 10e–g), but these were absent from the heads of SD7 refed flies (Supplementary Fig. 9b-e). Similarly, novel metabolic signatures were also present in the metabolic profiles of fasted SD fed flies (Supplementary Fig. 11a–d). It is particularly interesting to observe that some of these changes occurred after 2 days exposure to a SD and were maintained (Supplementary Fig. 10g), while others developed only at 7 days (Supplementary Fig. 10b).

To investigate whether the changes in metabolic signatures with short and longer exposure to the high sugar diet corresponded to transitions to distinct metabolic states, we conducted PCA in the refed and fasted conditions (Fig. 6b, c). We found that in each feeding state metabolites separated into four clusters defined by the time on the SD. Samples from SD2 and SD5 were largely overlapping, but entirely distinct from SD7 heads, suggesting that after long-term exposure to a high sugar diet, flies entered a different metabolic state. To identify which metabolites and pathways defined these new states, we examined the compounds that drive dominant patterns in PC1 and PC2 in the refed (Fig. 6d) and fasted (Fig. 6e) state. In the refed state, PC1 separated the metabolic profiles of SD7 heads, compared to other PCs. We used the compounds that contributed the most variation (PC1, top and bottom quartiles) for metabolic pathway network analysis. Compounds positively correlated with SD7 were enriched in aspartate metabolism ($p < 2.2 \times 10^{-4}$, FDR corrected hypergeometric test, SMPDB: SMP0000033), the urea cycle ($p < 2.4 \times 10^{-4}$, FDR corrected hypergeometric test, SMPDB: SMP0000059), TCA Cycle ($p < 3.7 \times 10^{-4}$, FDR corrected hypergeometric test, SMPDB: SMP0000057), glutamate metabolism ($p < 1.0 \times 10^{-3}$, hypergeometric test, SMPDB: SMP0000072) (Fig. 6d, Source Data). Methionine metabolism ($p < 4.9 \times 10^{-3}$, hypergeometric test, SMPDB: SMP0000033) and Phosphatidylcholine biosynthesis ($p < 2.7 \times 10^{-2}$, hypergeometric test, SMPDB: SMP0014306) were negatively correlated with the metabolic profiles of SD7 heads (Source Data). The fasted state datasets also clustered by PCA with time on diet, with SD2 and SD5 forming largely overlapping groups and CD and SD7 separating from the rest of the conditions (Fig. 6c). The

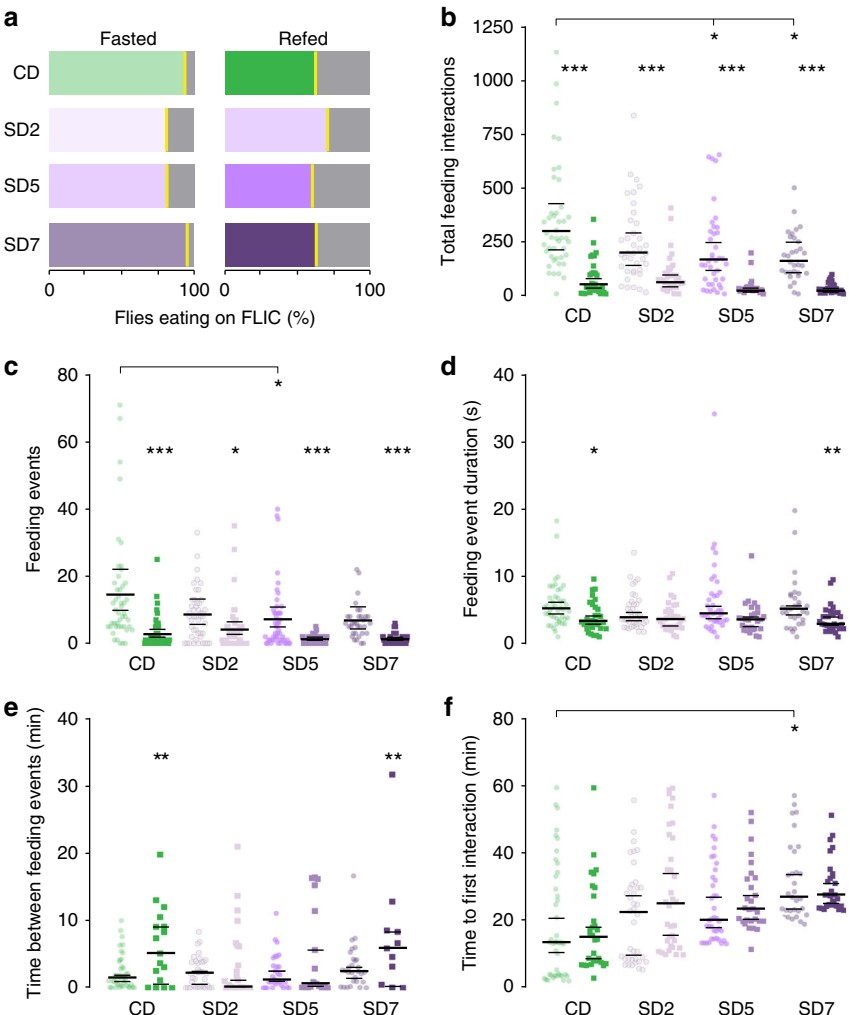

**Fig. 5** SD changes the magnitude of the behavioral responses to fasting. The feeding behavior of 24 h-fasted (lighter shades) and previously refed (darker shades) flies during ad-libitum access to 5% sucrose on the FLIC for 1 h. Control diet, CD (green shade) and high sugar diet day 2, 5, and 7 (SD2, SD5, SD7, purple shades), (See Supplementary Fig. 1 for a schematic). For panels **b**–**c**, which were analyzed using a Bayesian model (see Methods for details), bars show the median value and extent of a 95% central interval on the population-level mean for samples from the posterior predictive distribution; stars are assigned for each of several comparisons based on the posterior probability from Bayesian analysis of a difference in the indicated direction: ***$P > 0.999$, **$P > 0.99$, *$P > 0.95$. Comparisons are performed for each timepoint's fasted sample compared with the CD fasted sample, and for fasted vs. refed at each timepoint. For panels **d**–**f**, plotted bars show a bootstrap-based median and extent of a 95% confidence interval; **$p < 0.01$, *$p < 0.05$ using a stratified bootstrap; possible comparisons are identical to those in panels **b**, **c**. Fasted: CD $n = 44$, SD2 $n = 38$, SD5 $n = 39$, SD7 $n = 31$ biologically independent animals. Refed: CD $n = 30$, SD2 $n = 34$, SD5 $n = 28$, SD7 $n = 29$ biologically independent animals. **a** The percentage of fasted and refed flies that interacted with food on the FLIC (green or purple shades), compared to those that did not eat (gray shades). **b**–**f** The feeding behavior of flies during the 1-h ad-lib access to 5% sucrose quantified as **b** as the total number of feeding interactions; **c** the number of feeding events (defined as five of more consecutive feeding interactions above threshold); **d** the mean duration (sec) among feeding events; **e** the mean time between two or more feeding events (min); **f** the latency to interact with food (min), calculated as the first feeding interaction initiated on the FLIC. For panels **d**–**f**, replicates with no feeding events were excluded (for panels **b**, **c** the no-interaction events are explicitly accounted for in our zero-inflated negative binomial model). Source data are provided as a Source Data file

fasted state in a CD is defined by higher levels of NAA, kynurenine, aspartate, γ-glutamyllysine, and tyrosine (Fig. 6e), which are depleted by both short- and long-term exposure to SD (Supplementary Fig. 11). Thus, these compounds are not only uniquely sensitive to feeding state (Fig. 2), but also responsive to overall energy levels. Overall, our analysis identifies a number of signature metabolites and pathways that are changed with exposure to a high sugar diet; together, this provides a starting point to investigate potential connections between metabolites levels, inter-organ communication, complex behaviors, and dietary environment.

## Discussion

Metabolites are biologically active compounds that are more than just fuel for the body: they modulate cellular physiology and play a central role in health and disease. Since their influence on complex behaviors is unclear, we set out to study this question by first mapping the metabolic changes that underlie the transition between hunger and satiety, which is easy to quantify with behavioral assays. To do this, we developed a feeding protocol that resulted in rapid changes in the foraging and feeding behaviors of *Drosophila melanogaster* flies and used it to measure metabolites that change in the heads and bodies during the shift

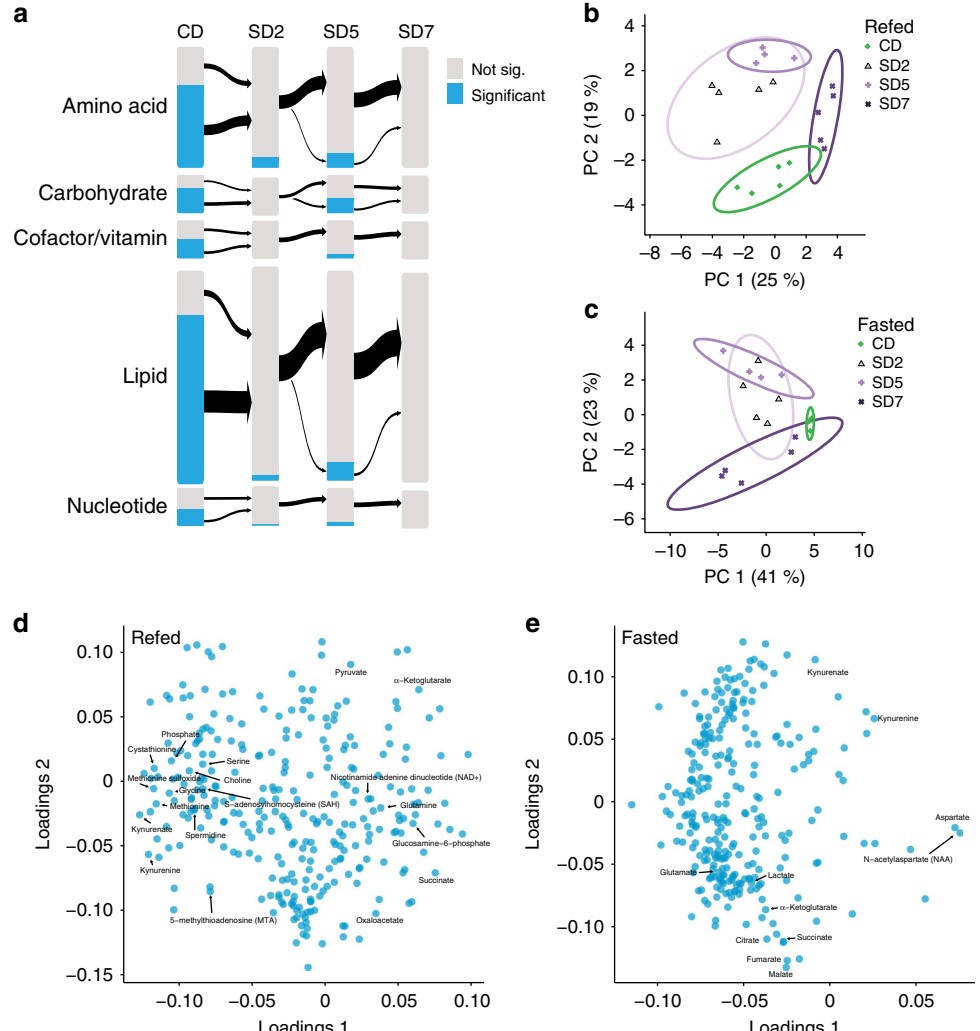

**Fig. 6** SD reshapes the metabolic transitions between hunger and satiety in fly heads. **a** The effects of short, (2 days, SD2), medium (5 days, SD5), and longer term (7 days, SD7) consumption of a high sugar diet (SD) compared to a control diet (CD) on the magnitude of the metabolite differences between the fasted and refed state on each diet. The size of the bars is proportional to the number of compounds in each metabolite category on the left. In blue are metabolites that were significantly different between fasted and refed flies in each condition (CD, SD2, SD5, and SD7 by Welch's $t$-test, FDR < 0.1), and in gray metabolites that were not. **b**, **c** Projection onto the two most explanatory principal components for compound levels from **b** refed and **c** fasted heads of flies on CD (green) and fed a SD (purple shades) for 2 days (SD2), 5 days (SD5), or 7 days (SD7). The solid lines represent the 95% confidence interval, and percent explained variance is listed in parentheses. **d**, **e** Principal component loadings for the **d** refed and **e** fasted heads of flies fed a CD, or SD2, SD5, or SD7. Labels represent characteristic compounds. Source data are provided as a Source Data file

between hunger and satiety. We also asked how short- and long-term consumption of a high sugar diet, which is known to promote obesity and alter feeding patterns in flies[14], changes acute behavioral and metabolic responses to fasting and refeeding. To aid the analysis of *D. melanogaster* metabolomics data, we developed Flyscape, an open-access application for Cytoscape where users can visualize and understand metabolomics data in the context of *D. melanogaster* networks.

We found that while metabolites change rapidly upon refeeding in both the head and body tissues, the responses of each to fasting and eating are distinct. Overall, heads show large fold increases in glycolytic, pentose-phosphate, 1-carbon metabolism, and hexosamine biosynthesis pathways intermediates that were largely absent in bodies, pointing to a faster response to nutrient availability. Consistent with this observation, we found that the RNA abundance of several nutrient transporters and metabolic enzymes also changes rapidly in the brains of fasted and refed flies. Of note, the levels of

biosynthetic precursors to neurotransmitters responded to feeding state only in head tissue even when present in both samples: choline, N-acetyl serotonin, glutamate, and GABA increased in refed flies, while aspartate and NAA were higher in fasted animals. Since these molecules have known or emerging roles in modulating cellular behavior, it is tempting to speculate that the changes we observe could have informational value and, thus, affect brain function. However, it is also possible that the behaviors with the fasted and refed states have little to do with metabolite levels, and are instead mediated by circuit and synaptic mechanisms that are modulated by hormone levels, neuropeptide signaling, and inter-organ communication. Nonetheless, considering that the internal energy state of animals influences many behaviors[15], including feeding[15,16,18,36], learning and memory[37], and sleep[38], our work on the identification of the metabolic signatures characterizing these states, opens the road to functionally test the role of metabolic signaling in behavior.

Consumption of a high sugar diet profoundly altered the metabolic profiles of fly bodies. Importantly, many of the metabolic hallmarks that characterize humans with obesity occurred in flies fed a high sugar diet, such as elevations in branched-chain amino acids, advanced-glycan products, glutamate, and α-ketoglutarate[28,32–34,39], are consistent with the findings that fruit flies fed a high sugar diet develop obesity-related illnesses[30,40]. Bodies also showed signs of TCA cycle dysfunction and a depletion in nucleotide metabolism, as previously observed in tissues of obese mammals[41] and humans[33,34]. These findings extend previous studies on the effect of a high sugar diet on the metabolism of fruit fly larvae[42,43] and provide an inroad to study the contributions of these metabolic changes contribute to obesity and metabolic disease in a genetically tractable model organism.

In both heads and bodies, a high sugar diet also led to a flattening of the difference in metabolite levels between the fasted and refed state. Given that fluctuations in metabolites critically control cell physiology through gene regulation, protein modification, and second messenger signaling[1–3], an enticing question is if and how this metabolic dulling impacts cell function and physiology. For example, we observed that in heads nearly all the fluctuations in the levels of neurotransmitters and their precursors between feeding states disappeared with both acute and long-term consumption of the high sugar diet, but whether these or other changes in metabolites levels with fasting and sugar diet exposure influence neuronal circuit function and behavior remains to be seen.

Metabolic remodeling has been widely studied in both stem cells and cancer and found to play a crucial role in cell physiology and disease progression[6,44]. To see if consumption of a high sugar diet also leads to a shift in head metabolic state, we examined the metabolic profiles of the heads of flies fed a high sugar diet for 2, 5 and 7 days. We found that longer exposure to this diet led to higher TCA cycle and hexosamine biosynthesis and lower glycolytic activity. The hexosamine biosynthesis pathway is considered a sensor of cellular sugar levels[45] and is involved in modulating feeding behavior[14,46,47], especially when animals eat a high sugar diet[14]. Neurons have high demand for cellular ATP for action potential generation and the restoration of membrane potential[48], but glycolysis for fuel is dispensable and inefficient in neurons[49–51], and instead used for cellular signaling events, such as those occurring in glucosensing neurons or synaptic plasticity. The high need for energy is met by glia, which metabolize glucose and trehalose into TCA cycle intermediates such as lactate and pyruvate, which are then transported into neurons to fuel cellular processes. The decrease in glycolytic and the increase in TCA intermediates we found in the heads of flies on a high sugar diet, raises the question of how these changes impact the proper functioning of neurons, especially in the contexts of cellular bioenergetics, neurodegenerative diseases, and information exchange. Exposure to a high sugar diet also depleted metabolites involved in 1-carbon metabolism. Given the relationship between 1-carbon metabolism, histone/DNA/RNA methylation levels, and gene expression[8,9], an exciting question is how changes in brain function and behavior with diet-induced obesity may be related to alterations in gene expression due to this reprogramming of metabolism. Finally, the metabolites NAA and kynurenine also showed depletion in the heads of flies on a high sugar diet. NAA, the second most abundant human brain metabolite, is lower in the brains of people with a variety of neuronal diseases and conditions, including depression, schizophrenia, dementia, Alzheimer's, stroke, and traumatic brain injury[52]. Like human brains[53], fly heads contain high levels of NAA; we found that these change with internal energy state and diet. In flies, high internal energy conditions, either due to the refed state or

consumption of a high sugar diet, had lower NAA, while fasting increased it. Interestingly, NAA levels are also lower in the brains of humans with obesity[39,54]. Thus, our data suggest the possibility that NAA is not only a sentinel of brain health, but also a marker of the overall brain energy state and point to a potential role for this metabolite in fueling cellular energetics in a stress or nutrient-deprived state[53]. Experiments that functionally address the effects of different NAA levels on brain physiology, will help elucidate the function of this metabolite. The changes in kynurenine levels with fasting and a high sugar diet we discovered are also worth noting. Kynurenine and kynurenic acid, the by-products of tryptophan degradation, are increased by exercise[55], decreased by chronic-stress and trauma[56], and were recently linked to the physiology of depression[56]. Together, the link between obesity and depression, and our data showing that kynurenine is correlated with brain energy state and feeding behavior, warrant a deeper investigation on the role of this metabolite in the fine-grained control of food intake, energy balance, and mood.

Overall, our data suggest that the internal energy state of the animal, whether it is fasting, satiety, or a high sugar diet alters the way in which nutrients are assigned to metabolic pathways. Metabolic adaptations to environmental and nutritional challenges vary depending on the tissue metabolic needs. While a few of these changes occur gradually and dull metabolite fluctuations between the fasted and fed states, our data show that most transitions are new and may reflect a passage to a new state, reminiscent of some sort of metabolic reprogramming. Mapping these metabolic transitions is the first step towards understanding their potential effects on the physiology of the brain and the role of metabolic signaling in the development of brain conditions associated with diet, such as neurodegeneration, depression, and seizures. In particular, by pinpointing the metabolites that characterize different internal energy states, our work has shed light on the types of metabolic information available to potentially modulate complex behaviors that change with internal energy, such as feeding, sleep, mood, and cognition. While our manuscript does not draw any causal connections between metabolites levels and complex behavior, our analysis provides a springboard to study the function of metabolites as messengers and transducers of environmental information in neuroscience.

## Methods

**Fly rearing and maintenance**. Flies were raised on Bloomington cornmeal medium (100% (v/v) water, 1.69% (w/v) brewer's yeast (MP Bio), 0.98% (w/v) soy flour (NutriSoy), 7.13% (w/v) yellow cornmeal (GFS), 0.56% (w/v) Drosophila Agar, Type II agar (Apex), 7.5% (v/v) white corn syrup (GFS), 0.86% (v/v) propionic acid (Fisher Scientific), and 0.4% (v/v) Tegosept (10%, mixed with 95% EtOH) (Apex)) inside incubators at 25 °C with a 12:12 dark:light cycle. At eclosure males were sorted into control diet and aged for 2–4 days. Age-matched males where then switched into vials containing a control diet (CD) or a diet supplemented with 30% sucrose (SD) as previously described[14,31]. Animals were provided fresh food every other day. All experiments used $w^{1118}CS$ male flies, given that their food preferences are not influenced by mating status[57].

**Fly-to-Liquid-Food Interaction Counter (FLIC) Feeding assay**. Thirty age-matched male 4–5-days-old flies were fed CD or a SD for 2, 5, or 7 days and then fasted for 24 h in vials containing a Kim-wipe imbibed with 2 mL of MilliQ water at ZT 5, and then transferred to vials containing 1% agar alone (fasted) or agar supplemented with 400 mM D-glucose (refed) for 1 h. The agar was colored with 0.5% FDC Blue to monitor feeding. Flies were anesthetized on ice and loaded onto the FLIC with an aspirator and given ad libitum access to 5% sucrose for 1 h. The individual feeding interactions were recorded five times per second with a signal threshold of 40, and analyzed using R with code developed by the Pletcher lab[13]. R code used to analyze raw FLIC data is included as an R markdown file (Supplementary Note 1) and raw data are included as Supplementary data 1. Only flies eating at least two events were used to calculate the "time between feeding events"; all other parameters were measured for flies with >0 feeding interactions, except the data in Fig. 1a and Fig. 5a which include all the flies, even those with feeding interactions = 0. Mean time in between feeding events only measured the duration

between feeding events, not the time before the first feeding event and the time between the last feeding event and the end of the recording (1 h). Feeding events and mean feeding event duration also exclude any events that are not concluded before the end of the recording period. An event is defined as five consecutive feeding interactions above the threshold.

Statistical analysis for the FLIC: Data on the number of feeding interactions and number of feeding events were fitted using a Bayesian zero-inflated negative binomial model using the R package brms[58], in order to properly account for both the count-based nature of the data and the presence of a large and variable number of no-interaction replicates. For each data set we applied a hierarchical model for the response variable (feeding events or feeding interactions, as appropriate) with population-level (fixed) effects for the treatment (diet and feeding/fasting) and group-level (random/batch) effects for the FLIC device. In all cases except for the data shown in Fig. 1b, we also found that the data justified fitting the zero-inflation term as a function of the treatment conditions. Models were fitted using brms[58] and convergence assessed using the Rhat criterion and manual inspection of posterior predictive distributions. We used default uninformative priors for all parameters, which involve uniform (on the real numbers) priors on the population-specific regression parameters themselves, a Student $t(3,4,10)$ prior on the (log-scaled) intercept term for the event rate, logistic$(0,1)$ priors on the zero-inflation terms, half-Student $t(3,0,10)$ priors on the standard deviation parameters, and a gamma$(0.01,0.01)$ prior on the negative binomial shape parameter.

For the mean duration, mean time, and latency statistics in Figs. 1, 5, statistics were assessed using bootstrapping with 1000 bootstrap replicates, with resampling stratified by FLIC device to control for batch effects. We used bias-corrected accelerated bootstrap estimates[59] for a 95% confidence interval on the median, and performed hypothesis tests against a null bootstrap distribution in which labels of the groups to be compared were randomized, using the difference in medians as a key statistic. Processed data are included as Supplementary Data 3.

**Foraging assay**. Fasted and refed flies were tested for their motivation to forage as in ref. [60]. Briefly, 24–28 age-matched male 3–4-day-old flies were fed a control or high sugar diet for 2, 5, or 7 days and then fasted for 24 h in vials containing a Kim-wipe imbibed with 2 mL of MilliQ water. The flies were then transferred to an arena containing a small food cup (1% sucrose agar colored with 0.5% FDC blue). After 1 h flies were anesthetized on ice and scored for the presence of blue food in their abdomens. Data were analyzed using GraphPad Prism 8.0.0.

**Triglyceride measurements**. Triglycerides were measured according to ref. [12]. Briefly, 24–28 age-matched male 3–4-day-old flies were fed a control or high sugar diet for 7 days and then fasted for 24 h. The flies were then transferred to vials containing 1% agar alone (fasted) or agar supplemented with 400 mM D-glucose (refed) for 1 h. The agar was colored with 0.5% FDC Blue to monitor feeding. Flies were then immediately collected into 1.5 mL microcentrifuge tubes and placed onto dry ice storage at −80 °C before conducting the assay. Data were analyzed using GraphPad Prism 8.0.0 using linear regression from known standards.

**Sample collection and processing for metabolomics analysis**. To achieve the two different feeding states, animals were fasted for 18–24 h in the presence of water. Animals were then refed agar (fasted) or agar supplemented with 400 mM D-glucose (refed) colored with a food dye to visualize feeding for 1 h. Animals were immediately sacrificed in liquid nitrogen and their heads and bodies collected by sieving. We opted to use heads rather than brains, because the dissection process introduced a large source of variability likely due to the slow quenching of metabolic processes. To collect samples from different diets, flies were maintained on a control cornmeal diet (~5% sucrose) or a high sugar diet (30% sucrose) for different days, but were all age-matched and collected at the same time. Each biological replicate consisted of 100 heads or 50 bodies; replicates were collected in parallel, stored at −80 °C and sent to Metabolon Inc. (Morrisville, NC) overnight for sample processing.

**Metabolomics sample preparation**. Samples were prepared using an automated MicroLab STAR system from Hamilton Company. Several recovery standards were added prior to the first step in the extraction process for quality control purposes. To remove protein, dissociate small molecules bound to protein or trapped in the precipitated protein matrix, and to recover chemically diverse metabolites, proteins were precipitated with methanol under vigorous shaking for 2 min (Glen Mills GenoGrinder 2000) followed by centrifugation. The resulting extract was divided into five fractions: two for analysis by two separate reverse phase (RP)/UPLC-MS/MS methods with positive ion mode electrospray ionization (ESI), one for analysis by RP/UPLC-MS/MS with negative ion mode ESI, one for analysis by HILIC/UPLC-MS/MS with negative ion mode ESI, and one sample was reserved for backup. Samples were placed briefly on a TurboVap (Zymark) to remove the organic solvent. The sample extracts were stored overnight under nitrogen before preparation for analysis.

Samples were run in a Waters ACQUITY ultra-performance liquid chromatography (UPLC) and a Thermo Scientific Q-Exactive high resolution/accurate mass spectrometer interfaced with a heated electrospray ionization (HESI-II) source and Orbitrap mass analyzer operated at 35,000 mass resolution. The

sample extract was dried then reconstituted in solvents compatible to each of the four methods. Each reconstitution solvent contained a series of standards at fixed concentrations to ensure injection and chromatographic consistency ("internal standards"). After reconstitution a small aliquot from each sample was combined into a pooled sample and run periodically over the duration of the experiment ("pooled biochemicals"). Metabolites were quantified as area under the curve and normalized to protein levels for each sample to account for potential differences in the amount of material present in each sample. Peaks were identified based on authenticated standards that contains the retention time/index (RI), mass to charge ratio (m/z), and chromatographic data (including MS/MS spectral data) on all molecules present in the library.

**Metabolomics data analysis**. All statistical analysis of metabolomics data was conducted in the R version of Metaboanalyst (R: 3.5.2, MetaboAnalystR: 1.0.2, Windows 10 × 86_64-w64-mingw32/x64 (64-bit), RStudio 1.1.414)[61]. Compounds with more than 50% missing values for each condition were removed and the remaining missing values were replaced by half of the minimum value. An RSD cutoff was not applied to the data, which may lead to false-negative observations when considering biochemicals that did not achieve significance. The data were then normalized using "Range scaling". Data were clustered by Ward's method (Euclidean distance metric) and plotted as heatmaps. The code used for statistical tests is included as an R markdown file (Supplementary Note 2). The transition plot was generated using R package Gmisc (Fig. 6b)[62].

**RNA sequencing and analysis**. Ten central brains of fasted, 2 h refed, or sated (collected after their natural morning meal) flies were rapidly dissected in hemolymph like saline under a dissection scope and immediately placed in Trizol reagent for each biological replicate. RNA was precipitated using isopropanol and analyzed for quality and concentration using a Bioanalyzer. RNA-sequencing libraries were prepared with the Nugen Ovation *Drosophila* kit (0350–32) and sequenced on an Illumina NextSeq-500 at the University of Michigan core facility. Sequencing reads were aligned with STAR[63] and differential abundance calls were made by DESeq2 (1.18.1, R version 3.4.3)[64]. DAVID (6.8) was used for Gene Ontology enrichment analysis[65]. The code used for statistical tests is included as an R markdown file (Supplementary Note 3). For consistency, RNA-sequencing data on high sugar diet flies[35] was reanalyzed by the same methods. These animals were on a diet with similar concentrations of sugar (40% vs. 38%) but a different amount of protein.

**Flyscape development**. Flyscape is functionally similar to Metscape[21]. It is also based on standard 3-tier architecture. The Presentation tier consists of the plug-in for Cytoscape[66]. The Logic tier is a standard web service that provides access to the data stored in our database and specific logic such as deriving the metabolic graphs, or performing enrichment analysis. The Data tier is represented by a SQL Server database that is accessed by the web service. The database is unique to Flyscape. It was built by parsing data from BioCyc database and integrating information from Flybase. The major objects in the database are Compounds, Reactions, Enzymes, Genes, Pathways and their relationships. The Logic Tier web service provides a REST based interface. The data are returned in a standardized XML format that is easily translated into Java Objects in both the Presentation (Flyscape plug-in) and Logic tiers. Using this combination of open standards (REST/XML) allows for easy extension of the service.

Flyscape supports four types of network graphs: (1) compound (C), (2) compound-reaction (CR), (3) compound-gene (CG), and (4) compound-reaction-enzyme-gene (CREG). In CREG networks a metabolite node is connected to all reaction nodes that involve that particular metabolite and a reaction node is connected to all enzymes that catalyze this reaction and to all metabolites (substrates and products) involved in that reaction. An enzyme node, in turn, is connected to all genes that are known to encode that enzyme and to all reactions catalyzed by a given enzyme, and a gene node is connected to all enzymes encoded by that gene. Thus, nodes of the same types are never connected to each other. Edges between compound and reaction nodes are directional to signify the reaction substrates and products. The CG network is essentially a simplified version of the CREG network where reaction and enzyme nodes are not shown. Flyscape supports two types of queries. The user can: (i) supply a list of genes and/or compounds; or (ii) select one of the canonical metabolic pathways.

**Reporting summary**. Further information on research design is available in the Nature Research Reporting Summary linked to this article.

## Data availability
Raw FLIC data are in Supplementary Data 4. All raw metabolomics data used in this manuscript are in Supplementary Data 1. The raw data for the metabolomics experiments were deposited in MetaboLights[67] under accession number MTBLS180. Flyscape has been uploaded to the Cytoscape App Store (http://apps.cytoscape.org/apps/flyscape). RNA-sequencing reads and count were deposited in Gene Expression Omnibus accession number GSE126494.

The source data underlying Figs. 1a–f, 2a–e, 3a–d, 4a–f, 5a–f, 6a–e, and Supplementary Figs. 1b–c, 4a–f, 5, 6b–e, 7a–b, 8a–b, 9b–c, 10a–g, 11a–d are provided as a Source Data file. All other data are available from the corresponding authors on reasonable request.

## Code availability

All code used to analyze data is publicly available and listed below: The code for the statistical tests of the metabolomics data is included as an R markdown file (Supplementary Note 2) and also available as described in ref. [61] (R: 3.4.3, MetaboAnalystR: 0.0.0.9000, Linux 4.4.0–17134-Microsoft x86_64). For RNASeq, sequencing reads were aligned with STAR[63] and differential abundance calls were made by DESeq2 (1.18.1, R version 3.4.3)[64]. DAVID (6.8) was used for Gene Ontology enrichment analysis[65]. The code used for statistical tests is included as an R markdown file (Supplementary Note 3). R code used to analyze raw FLIC data is included as an R markdown file (Supplementary Note 1) and available at flidea (https://www.flidea.tech/projects). Flyscape is available at Cytoscape App Store (http://apps.cytoscape.org/apps/flyscape). The Flyscape code is available upon request.

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

## Acknowledgements

We thank Julia Kuhl for drawing the fruit fly graphics and Paul Trombley for designing the Flyscape logo. This work was supported by NIH R00 DK-97141 (MD), NIH 1DP2DK-113750 (MD), The Rita Allen Foundation (MD), NIH R35GM128637 (PLF), and T32 DA007268 (DW). The development of Flyscape was supported by the NIH Metabolomics Research Consortium Large Pilot/Feasibility grant U24 DK097153 (MD) and Michigan Metabolism and Obesity Center Large Pilot/Feasibility grant P30 DK089503–06 (MD).

## Author contributions

D.W. analyzed the metabolomics and RNA-sequencing data and assisted with the creation of Flyscape. J.W. performed the behavioral experiments and TAG measurements. W.D., J.M. and A.K. created Flyscape. J.L.P. collected the samples for metabolomics experiments. J.M.K. obtained the mass spectrometry data from the metabolomics samples. K.J.H. collected samples for the RNA-sequencing experiment. D.W., P.L.F. and M.K. analyzed the RNA-sequencing data. P.L.F. conducted the statistics for Figs. 1 and 5. M.D. and D.W. wrote the manuscript with input from all authors.

## Additional information

**Competing interests:** The authors declare no competing interests.

