## [Peer Review File · Nature Communications]

Reviewers' comments:

Reviewer #1 (Remarks to the Author):

The authors use a metabolomics approach to understand how nutrient starvation influences metabolism of the fruit fly, *Drosophila melanogaster*. The study focuses on two conditions. First, the authors examine how starvation affects metabolites levels in both the body and heads of adult flies. Secondly, the authors examine how exposure to a high sugar diet alters metabolism following a bout of starvation. While the study is largely descriptive, it will be highly cited and is important for two reasons: (1) Metabolomic analysis of starved fly heads provides an snapshot of how brain metabolism reacts to nutrient stress. To the best of my knowledge, this study is unique in that it examines how starvation and exposure to a high sugar diet influences brain chemistry in a genetically tractable model organism. (2) The development of Flyscape provides a powerful tool for *Drosophila* researchers. This application fills a major need in the community and will be widely used. My only criticism is that Flyscape is based on the *Drosophila* Biocyc database, which, while adequate, is still a work in progress and is clearly missing key enzymatic steps in a number of pathways; however, the issue can easily be addressed in future updates. Considering that this study represents the first attempt to adapt the powerful Metscape tool to flies research...the advance is in and of itself is worthy of publication.

I have no major concerns with this study, but have two suggestions that would improve the manuscript:

1. On a few occasions, the authors imply that changes in certain metabolite pools imply increased metabolic flux. For example, lines 162-164 state: "In our data, it reveals a change in carbon utilization in the TCA cycle..." Also, in lines 189-192, the authors state: "... argue that carbon unutilization [sic] into the TCA cycls..." Since the authors only measured steady state metabolite levels, these statements are simply speculation. Although I understand the temptation of making such claims, the authors should modify or remove any statements regarding changes in metabolic flux that require verification by stable isotope tracer analysis. Such studies are not necessary for publishing this manuscript and I encourage the authors to edit the relevant text.

2. The average *Drosophila* geneticist will find large sections of the text quite intimidating because of the large number of metabolic pathways referenced in the text. While the manuscript illustrates metabolic pathways in the supplemental figures, these schematic diagrams are a bit hard to follow. When possible, I'd encourage the authors to either provide illustrations of key metabolic pathways in the supplement OR provide a reference to the appropriate online KEGG metabolic pathway diagram.

Reviewer #2 (Remarks to the Author):

In this manuscript, Wilinski et. al. measured the metabolic profiles in *Drosophila* (head and body) at several different feeding and nutrition conditions, and tried to correlate the changes in metabolic profiles with feeding states and diets. Overall, this work provided some potentially interesting metabolomics data (and part of RNA data) to the field. However, I feel no conceptual advance was achieved in this work. More important, all conclusions in this work are descriptive (based on correlative analyses and literature reviewing), no biological validations were performed. Therefore, I think this work should be not considered for publication at the present form.

Specific comments for technologies and data quality:

1. The authors claimed that they used a large-scale metabolomics profiling (provided by Metabolon Inc) for this work. However, only 391 metabolites were measure in this study, while some of them are lipids (85). In another word, only ~300 metabolites were measured, which is a small fraction compared to a total of 4856 metabolites in Flyscape. The coverage of lipid analysis cannot be comparable to the lipidomics. Therefore, in whatever standard, this work should not be called “large-scale” metabolomics. The authors should increase the profiling coverage and add lipidomics into the analysis.
2. Experimental design and data quality: in most groups, only 3-5 biological replicates were employed. Given the relatively easy effort to obtain the fly samples, and the large variations for in vivo sampling, more biological replicates are generally required for metabolic profiling for in vivo studies, usually 6-10 biological replicates in each group.
3. No data in the manuscript was provided to characterize the data quality and measurement reproducibility. For example, in PCA plots (Figure S1, and Figure 6), no QC samples were used to monitor the analysis reproducibility. No RSDs were calculated to indicate the quality of metabolite measurements. In addition, the data distributions in Figure 6 and Figure S1 are relatively large, which might indicate the poor reproducibility. But without the data quality assessments, I cannot evaluate this point. Please add more data quality assessments.
4. Data analysis: the authors did not clearly explain the quantitative conditions for “changed metabolites” (such as Page 5, Line 100; Page 12, line 259-260). Only in Page 8, line 174, the authors

state that “Of the 391 compounds measured, 185 changed (t-test, FDR < 0.1)...” . I guess this is p value with FDR correction<0.1. If so, I think the condition is too loose for quantitative analysis (at least 0.05). The loose cut-off values may be also contributed from the small number of biological replicates or poor data quality (see comments 2 and 3).

5. Flyscape is a good bioinformatic tool, but the general concept follows the previous publication (Metscape, ref 17). No major improvement was made to Flyscape. Only BioCyc database was used for Flyscape. How about the KEGG? How many of 391 metabolites are mapped to Flyscape? I think if the KEGG could be added to Flyscape, the coverage should be further improved.

Specific comments for biological part:

6. The authors designed several feeding experiments to mimic different physiological/pathological conditions. But NO biological data was provided to characterize the flies (e.g., health status and behaviors). For example, after the 18-hour starvation, *Drosophila* usually underwent stress, and the autophagy may also be activated. The measured metabolic profiles indicated the response to stress (at least in part), instead of the effect of “refed”. Similar issues are with other designed experiments. Only experimental conditions are described, no biological data was provided to prove that the animal models are normal, and truly related to the aimed physiological/pathological conditions.

7. Most conclusions in the manuscript were based on correlative analysis and literature reviewing. NO biological validated were provided. I hope the authors would validate at least one or two points through biological experiments.

After the data quality is improved, I can further evaluate the biological significance of the paper.

Reviewer #3 (Remarks to the Author):

Most experimental approaches in biology rely on recording the reaction of biological systems to experimental perturbations. This approach is especially relevant in “omics” approaches where a very large repertoire of parameters is recorded to measure the impact of experimental manipulations. These data are then used to reconstruct the rules or mechanisms governing the regulation of the interrogated biological system. Such approaches have been very powerful in many systems including *Drosophila melanogaster*. Given technological advances in instrumentation and bioinformatics, an

approach gaining popularity is the comprehensive survey of metabolites in cells and organisms. This so-called metabolome is however highly complex and notoriously difficult to interpret. Nevertheless, the current rebirth of physiology and metabolic research and the realization that nutrition is key to our understanding of many diseases, makes the measurement and interpretation of metabolome information extremely relevant. In this context, it is curious that despite the track record of the *Drosophila* community in trailblazing the development and deployment of new methods, the tools available to the fly community to analyze and interpret metabolome information are very limited. Especially when it comes to integrating information from metabolomics and other “omics” approaches. Also, the number of metabolomics studies in the fly is (still) limited. This is the premise on which the work described in this manuscript is based.

In the present manuscript, the authors describe a set of experiments in which they modify the availability of sugar in the diet of flies followed by metabolomics measurements and sometimes transcriptomics. They then use a newly developed tool to visualize their results in the context of largely already published transcriptomics data. The topic of the manuscript is therefore very timely and relevant for the *Drosophila* and metabolomics community. I have however some reservations about the overall structure of the manuscript and how some of the experiments have been designed and interpreted.

My main concern relates to how the work itself and the results are presented in the paper and how some analyses were performed. In general, I had the feeling that the authors often distort or exaggerate the impact of their study and what can be interpreted. For example, the title gives the impression that multiple dietary components are tested (sugars, vitamins, fatty acids, amino acids, nucleotides etc.) when in reality the authors only manipulate the sugar content of the diet. Also, the authors talk about feeding states in the title and it is not clear to me what they mean by that as they never quantify nor carefully look at feeding behavior in the animal. From performing a quick literature search it became apparent that some of the dietary manipulations used in the paper can lead to different changes in feeding behavior which seem to differ depending on the lab in which they are performed. This could be an additional concern and the authors should mention this and discuss the relevant papers. After reading the title and reading the abstract, reading the manuscript was a bit anticlimactic (and not because it is worse than suggested but because it is different to what was promised at the beginning).

I also found the design of the dietary manipulations a bit confusing. This then extends to the corresponding interpretation of the analysis of the metabolomics results. The authors perform a large set of experiments and then try to make generalizable interpretations. However, I must say that for me the different experiments are not easy to compare and do not help to generate a coherent picture. It is also not always clear to me why the authors first feed the fly, then starve it and then refeed it. I found that this complex design, which is augmented by the multitude of manipulations of the sugar content of the diet, leads to more confusion than clarity. Focusing on a specific manipulation or two and then diving deep into the interpretation of that one would have satisfied me and maybe allowed the authors to properly distill a clear interpretation and finding. The confusion is maybe augmented by a generally superficial and poor description of the content of the paper at the start of the results section. I would find it helpful to have a more thorough explanation. It was not always clear for me what conditions were tested, what was the origin of the data, and for which purpose the experiments were performed.

An additional important problem I have with the design of the experiment is that the authors often rely on published transcriptomics data to compare the effects they see in their metabolomics measurements with possible changes in the transcriptome in the fly. First of all, it took me quite some time to find out which data were generated by the authors and which were extracted from databases. Second, metabolomics should be very sensitive to differences in diet and other experimental parameters. To which extent are the diets and the experimental manipulations in the present study and the studies in which the transcriptomics data were generated comparable? Given the provided evidence I have no way to judge this (the authors do not even provide the detailed composition of the medium for the animals). I am therefore skeptical to which extent the presented metabolomics data can be compared with the transcriptomics data.

Furthermore, I am not sure that the authors make a good case about the usefulness of FlyScape. When going through the manuscript I had the impression that most of the findings described by the authors could have been achieved by simply analyzing the metabolomics and the transcriptomics results separately. I am sure that FlyScape can be a powerful tool but I have some reservations: From what I can see given the description of the FlyScape tool it does not perform any statistical analysis of the data like pathway enrichment etc. but uses precomputed statistical values. It is, therefore, a pure visualization tool. While this is useful I am not sure that I am convinced by the examples provided by the authors.

Finally, I am not convinced by all the specific findings the authors describe in the manuscripts. Especially when it comes to interpreting the meaning of the changes in specific metabolites. There are so many examples described and so many correlations made with papers (mainly from the vertebrate literature) that I am left to wonder how much I really learn from it. Often the overall findings feel trivial. Given that nothing is explored in depth and no follow-up experiments are performed my impression is that if you measure enough metabolites you will always find specific ones changing with your manipulation. And often these metabolites have been proposed to correlate with disease or condition x in a different system. Giving 1 or 2 examples would be enough. But only if the authors make a case of why this is relevant and how this can lead to important insights using the fly model. At this stage, it is just a long list of correlations, comparisons, and nothing more.

In short, I think the paper is very relevant and timely but the paper tries to be too many things at the same time: a description of a new bioinformatics tool, an in-depth analysis of metabolomics findings, and an exhaustive description of different experimental manipulations. At the end, as a reader, I could not see the forest from the many trees and the resulting superficiality makes it very difficult for me to evaluate in depth what the authors have done and how well they have done it. I am convinced that if the authors focus on a limited set of messages, focus on a specific small set of manipulations and then dive deeply into them this will result in a clear and enriching manuscript. I general like the work and I am quite sure that FlyScape could be a useful tool but at this stage, I find it not suitable for publication in its current form.

Minor comments:

- I would plot the number of metabolites changing in a specific class as % and not the total number. It could be that for example many lipids are found changing because the measurement method is really good at detecting lipids.
- I would recommend that the authors try to avoid hyperbole like “unparalleled resource” (line 76).
- Given that glia play a key role in the metabolic dynamics of the brain the authors should give that interpretation of their results and the related literature (especially in *Drosophila*) more attention. Especially for the finding described in lines 190-92.
- The title in line 207 is highly trivial.
- There are so many comparisons in the paper that I often lost track what the authors are comparing to what (line 219 for example).
- Feeding state and nutrient state are used a lot and it is not clear what the authors mean by this. They should avoid this term or clarify it.
- The numbers of metabolites in the text and the figure sometimes do not match (Figure 1).
- The statements in line 421-423 and the end of the discussion are overreaching quite a bit. The experiments on their own are interesting enough and the physiology of animals fascinating enough. Except if the authors can be specific those parts are just marketing exaggerations.
- The authors should include the detailed composition of the so-called “Bloomington recipe” and also describe carefully how the transcriptomics experiments they used from the literature were done and how those conditions compare to the metabolomics ones.
- The colors used in the figures are often very similar and it is difficult to differentiate the different metabolite classes in the plots.

Dear Reviewers,

We would like to convey our thanks for your insightful and constructive comments. We were pleased to see that all three reviewers expressed excitement for our work and commented on the potential impact of Flyscape to the study of *Drosophila melanogaster* metabolism, as **Rev. 1** wrote: *“To the best of my knowledge, this study is unique in that it examines how starvation and exposure to a high sugar diet influences brain chemistry in a genetically tractable model organism. The development of Flyscape provides a powerful tool for Drosophila researchers. This application fills a major need in the community and will be widely used,”* and **Rev. 3** stated *“The topic of the manuscript is therefore very timely and relevant for the Drosophila and metabolomics community,”* and **Rev. 2** *“Overall, this work provided some potentially interesting metabolomics data (and part of RNA data) to the field.”*

You also raised several concerns about our study, mostly revolving around the **lack of clarity** and **focus** in our work. At the reviewers’ advice, we have performed a major revision of our manuscript by adding behavioral experiments, reformatting the figures, and rewriting a large part of the text. **We have outlined below the major changes and additions to the manuscript, before addressing the reviewers’ comments individually.** We strongly feel that these changes provide focus and clarity to the manuscript and better frame the rationale and the impact of our work.

Outline of changes to the revised manuscript:

1. Changes to the figures and text to clarify findings and make them more accessible
2. Addition of experiments to characterize the feeding behaviors and physiology of flies
3. Expanded Method session with more detail on data acquisition and analysis

1. Changes to the figures and text to clarify findings and make them more accessible.

We have changed the figures as outlined below and performed a substantial rewrite of the manuscript, including the abstract, introduction, and discussion. In this version we devoted space to explain the rationale for our experimental approach and focused the discussion on a few of the key findings. Briefly, as we now explain in the introduction, we were interested in **understanding the role of metabolites in modulating complex behavior.** As a starting point to

tackle this question, we decided to map the metabolic change that occur during the transition between hunger and satiety in flies. Indeed, while we know that behaviors -- from food intake to food choice to sleep -- change depending on the internal energy state of the animal (hunger or satiety), our knowledge of the metabolic changes that occur during this shift is poor. To begin addressing this question, we designed feeding manipulations that, as described below and in Fig. 1, result in a fast behavioral transition between hunger and satiety states. This allowed us to use these same conditions for metabolomics analysis, which led us to **identify the metabolites that change as the behavioral state of the fly is transitioning between hunger and satiety.**

We also decided to investigate how high nutrient diets change these metabolic profiles because we and others have observed changes in behaviors that depend on the internal state when animals are fed high nutrient diets. Such diets are known to reshape cell physiology, but most studies have largely been done in the context of immunology, cancer, and developmental biology, not neuroscience. Thus, pinpointing how a high sugar diet alters metabolic profiles associated with a change in the internal state is a first step to understand their effect on physiology. Our analysis uncovered ~20 metabolites and a handful of pathways that characterize the fasted and refed state on the two diets, which provides a molecular handle to address the function of each of these pathways in altering behaviors such as sleep, feeding, and memory (which change with internal state) in animals fed a high sugar diet.

In the new version of our manuscript, Figures 1 characterizes the behavior of hungry and refed flies. Figures 2 and 3 show the compounds changed in hungry and refed flies and how these differ in the heads and bodies. Figures 4-6 focus on the high sugar diet, and highlight its effect on body metabolism and feeding behaviors (Fig. 4 and 5) and identifying the metabolic signatures of the heads of fasted and refed flies fed a sugar diet (Fig. 6).

Outline of figure changes

Fig.1 is new and contains a characterization of the feeding behaviors of fasted and refed flies.

- Suppl. Fig. 1 is new, it shows how the foraging behavior and the triglycerides are affected by refeeding in animals on a control or sugar diet.
- Suppl. Fig. 2 shows the Flyscape workflow. This was originally Fig. 2.
- Fig. 2 now shows the metabolites that change in the bodies and heads of fasted and refed flies, and the application of Flyscape visualization to a subset of these data. These data

were originally in Fig. 1, Fig. 2 and Fig. 3 and Suppl. Fig. 2a. We removed the pie charts at the suggestion of reviewers.

- Suppl. Fig. 2 is new shows how all the metabolomics samples and the pooled samples cluster.
- Suppl. Fig. 4 show the PCA analysis for all the metabolomics conditions in this manuscript, this was originally Suppl. Fig. 1.
- Suppl. Fig. 5 shows the changes to neurotransmitter metabolism, it was originally Suppl. Fig. 3.
- Fig. 3 now shows the changes in RNA abundance in the brains of fasted vs. refed and fasted vs. sated flies and an application of the multi-omic data integration feature of Flyscape. We reformatted the data visualization for clarity. A subset of these data, visualized in a different format, was originally in Fig. 4.
 - Suppl. Fig. 6 shows our analysis of the brain RNAseq data, was originally Suppl. Fig. 4.
- Fig. 4 now shows the effects of a high sugar diet on the fasted and refed metabolome of bodies. At the suggestion of reviewers, we changed the way these data are visualized. These data was originally in Fig. 5.
 - Suppl. Fig. 7 shows the effects of diet in fasted and refed fly bodies, different pairwise comparison than in Fig. 4 focusing on diet effect on the same feeding state, was originally Suppl. Fig. 6.
 - Suppl. Fig. 8 shows the the changes in distribution of the metabolomics data, it was originally Suppl. Fig. 5.
 - Suppl. Fig. 9 shows our analysis of the RNAseq data from flies on a sugar diet from another study, it was originally Suppl. Fig. 7.
- Fig. 5 is new and now shows the effects of a high sugar diet (days 0-7; flies are age matched) on the feeding behaviors of fasted and refed flies.
- Fig. 6 stayed mostly the same (we removed the schematic in panel a), but has the addition of panels f and g which show correlations between the levels of two metabolites and the behaviors of fasted flies.
 - Suppl. Fig. 10 and 11 show metabolites that change with refeeding (Fig. 10) and fasting (Fig. 11) when animals are on a high sugar diet for days 2, 5, 7 compared

to animals on a control diet. The panels on the right of the heatmaps were added to more easily visualize the changes at the reviewers' suggestion. These were originally Suppl. Figs. 8 and 9.

2. Addition of experiments to characterize the feeding behaviors and physiology of flies.

Reviewers 2 and 3 commented that while we discussed throughout the paper the fasted and refed states, we never showed any data that these conditions actually reflected changes in the behavior of flies. This is an excellent point: we should have provided a context for our conditions. To this end, we added a full behavioral characterization of the fasted and refed conditions by doing these experiments:

- a. **High-resolution behavioral measurements of the feeding behaviors of the refed and fasted flies fed a control and sugar diet for 2, 5, and 7 days using the Fly-to-Liquid-Food Interaction Counter (FLIC)¹.**
- b. **An analysis of the foraging behavior of fasted and refed flies fed a control and sugar diet for 2, 5, and 7 days diet.**
- c. **Measurements of triglycerides in fasted and refed flies fed a control and sugar diet**

In the new Figure 1 and Suppl. Figure 1 we show that flies in the fasted and refed conditions have fundamentally different feeding behaviors and motivation to forage for food. Fasted flies find a hidden food source more quickly and eat more, while refed flies forage and eat less. Thus, providing a meal of 400 mM D-glucose for 1 hr is sufficient to alter the internal state of flies from hungry to sated. When on a high sugar diet (Fig. 5), flies do not show a decrease in their motivation to look for food, but the differences in the feeding behaviors between the sated and fasted states decreases.

3. Expanded Method section with more detail on data acquisition and analysis

To make the analysis of the behavior, metabolomics, and RNA-sequencing data more transparent and accessible to the public, we have created R markdown documents for each analysis and included tables of processed data (Supplemental Texts). The raw RNA-sequencing

data has been deposited in GEO, and included the raw FLIC and metabolomics. We have also included additional details on the acquisition of the metabolomics data to the Methods.

Response to Reviewer-specific comments

Response to Reviewer 1 comments

We thank reviewer 1 for the pointing out the strengths of our manuscript. We agree that the conditions tested (fasted and refed flies) provide a context for studying of organisms respond to changes in nutrient availability, and that Flyscape will be a useful tool for the community, as they state:

*“The authors use a metabolomics approach to understand how nutrient starvation influences metabolism of the fruit fly, *Drosophila melanogaster*. The study focuses on two conditions. First, the authors examine how starvation affects metabolites levels in both the body and heads of adult flies. Secondly, the authors examine how exposure to a high sugar diet alters metabolism following a bout of starvation. While the study is largely descriptive, it will be highly cited and is important for two reasons: (1) Metabolomic analysis of starved fly heads provides an snapshot of how brain metabolism reacts to nutrient stress. To the best of my knowledge, this study is unique in that it examines how starvation and exposure to a high sugar diet influences brain chemistry in a genetically tractable model organism. (2) The development of Flyscape provides a powerful tool for *Drosophila* researchers. This application fills a major need in the community and will be widely used. My only criticism is that Flyscape is based on the *Drosophila* Biocyc database, which, while adequate, is still a work in progress and is clearly missing key enzymatic steps in a number of pathways; however, the issue can easily be addressed in future updates. Considering that this study represents the first attempt to adapt the powerful Metscape tool to flies research...the advance is in and of itself is worthy of publication. “*

We also thank reviewer 1 for suggesting ways to improve our manuscript:

1. *“On a few occasions, the authors imply that changes in certain metabolite pools imply increased metabolic flux. For example, lines 162-164 state: “In our data, it reveals a change in carbon utilization in the TCA cycle...” Also, in lines 189-192, the authors state: “... argue that carbon utilization [sic] into the TCA cycls...” Since the authors only measured steady state metabolite levels, these statements are simply speculation. Although I understand the temptation of making such claims, the authors should modify or remove any statements regarding changes in metabolic flux that require verification by stable isotope tracer analysis. Such studies are not necessary for publishing this manuscript and I encourage the authors to edit the relevant text”*

2. *“The average *Drosophila* geneticist will find large sections of the text quite intimidating because of the large number of metabolic pathways referenced in the text. While the manuscript illustrates metabolic pathways in the supplemental figures, these schematic diagrams are a bit*

hard to follow. When possible, I'd encourage the authors to either provide illustrations of key metabolic pathways in the supplement OR provide a reference to the appropriate online KEGG metabolic pathway diagram."

We have removed all instances where flux was mentioned and also added references to KEGG pathways in the text. We agree that as Flybase annotations improve, the usefulness and power of Flyscape will too.

Response to Reviewer 2 comments:

1. *"The authors claimed that they used a large-scale metabolomics profiling (provided by Metabolon Inc) for this work. However, only 391 metabolites were measure in this study, while some of them are lipids (85). In another word, only ~300 metabolites were measured, which is a small fraction compared to a total of 4856 metabolites in Flyscape. The coverage of lipid analysis cannot be comparable to the lipidomics. Therefore, in whatever standard, this work should not be called "large-scale" metabolomics. The authors should increase the profiling coverage and add lipidomics into the analysis."*

We appreciate the reviewer's concern here, and we certainly would not want to overstate the impact of our study. To the best of our knowledge, however, the number of metabolites measured in our study is the largest reported to date in *Drosophila*. For comparative purposes Bratty *et al.*² measured 55 metabolites, Chintapalli *et al.*³ 242 in whole flies, and Tennessen *et al.*⁴ measured 114 metabolites. In addition, the average number of identified compounds in publicly available metabolomics studies uploaded to the Metabolomics Workbench is 116⁵. So, while it is true that the number of metabolites we measured (391) is small compared to the total number of metabolites found in the Flyscape database (which reflects a complete metabolic reconstruction and thus a purely theoretical upper bound based on our current state of knowledge), it is also substantially larger than the sets measured by comparable experimental studies. Either way, we have removed any descriptions of our work as "large-scale" in the new version of the manuscript.

The disconnect between the metabolome generated via metabolic reconstruction (see *Paley and Karp*⁶ for details) and the measurable metabolome and largely due to the fact that metabolites

belong to a broad range of chemical classes and there is no single method to measure all metabolites reliably at the same time. The Metabolon platform used in our study is one of the best in the world with respect to metabolite coverage, which is why we selected it (despite the high cost) and likely the reason we were able to recover such a high number of metabolites (again, compared with what can be practically accomplished, rather than what might be hoped for based on purely theoretical concerns). In fact, we did perform preliminary experiments with core facilities and collaborators using the same samples, but Metabolon gave far superior coverage and technical consistency. We agree with the reviewer that broader coverage of *Drosophila* lipidome would be of great interest; however, lipidomic analysis has a large number of unaddressed challenges with both measurement, detection and analysis, especially in *Drosophila* where pathways are even more poorly mapped than mammals, which is why we feel this is beyond the scope of the present study.

2-3 “*Experimental design and data quality: Given the relatively easy effort to obtain the fly samples, and the large variations for in vivo sampling, more biological replicates are generally required for metabolic profiling for in vivo studies, usually 6-10 biological replicates in each group,*”

and

“No data in the manuscript was provided to characterize the data quality and measurement reproducibility. For example, in PCA plots (Figure S1, and Figure 6), no QC samples were used to monitor the analysis reproducibility. No RSDs were calculated to indicate the quality of metabolite measurements. In addition, the data distributions in Figure 6 and Figure S1 are relatively large, which might indicate the poor reproducibility. But without the data quality assessments, I cannot evaluate this point. Please add more data quality assessments.”

Out of the 12 conditions examined in the current study, **9 conditions have n=5 biological replicates**, **2 conditions have n=4 biological replicates** (one sample in each was removed as an outlier), and only **1 condition has n=3** (a sample was lost during processing and one was an outlier). Each of the samples originated from pools of 100 fly heads or 50 bodies. The number of samples used in previously published *Drosophila* metabolomics studies ranged from 4 to 6 with each sample containing 10-20 animals (larvae in the majority of studies). Thus, our study falls within the range of standard numbers of biological replicates, while using more animals per replicate (which adds robustness). We also note that Metabolon adds internal standards to each

sample prior to injection into the mass spectrometers to calculate instrument variability, and runs pooled samples to calculate processing variability. In our samples, the median relative standard deviation (RSD) for the instrument was 3% and for the pooled samples was 8%. See Supplementary Fig. 2 for a PCA plot of the data and control pooled samples. Both of these RSD figures are considered quite low in the field and pass the quality control set by Metabolon. The reviewer is concerned about the widths of the distributions in Figs. 6 and Supplemental Figure 3; what is shown here is the result of an unsupervised data projection method, not an optimized method for obtaining a clean separation between the data sets. The lines represent the 95% confidence intervals. Even so, obvious separations between the different conditions are observed in nearly all cases, with the only exceptions being cases where we expect (and indeed argue) that fasted and refed conditions have blunted an otherwise-expected change in metabolic state.

To confirm that our experiments match the best practices of the metabolomics community, we examined the metabolites' RSD among samples under each experimental condition. The median RSD ranged from 27 to 48 (average of medians for each condition is 35.2%). While the vast majority of *Drosophila* metabolomics studies do not report an RSD and don't make their data publicly available so that it could be calculated, *Tennesen et al.*⁴ quote a median 49.9% RSD (and the use of 3-4 replicates per condition) and *Chintapalli et al.*³ report a RSD between 22-38% for "major" metabolites in two tissues, but do not report an overall RSD (that study used 4 replicates per condition). The performance of our analysis is thus uniformly consistent with best practices in the field. We also examined a number of individual metabolites that had higher RSDs and found that many such compounds are notoriously difficult to detect. One example is the TCA cycle metabolite pyruvate. We believe that in this and many other examples the higher RSD can be explained by the fact that a compound is co-eluted with many other compounds. In such cases the analysis software does not always integrate the peaks consistently.

4. *"Data analysis: the authors did not clearly explain the quantitative conditions for "changed metabolites" (such as Page 5, Line 100; Page 12, line 259-260). Only in Page 8, line 174, the authors state that "Of the 391 compounds measured, 185 changed (t-test, FDR < 0.1)...." . I guess this is p value with FDR correction<0.1. If so, I think the condition is too loose for*

quantitative analysis (at least 0.05). The loose cut-off values may be also contributed from the small number of biological replicates or poor data quality (see comments 2 and 3)."

We appreciate the reviewer's suggestion to increase the stringency of our FDR, although we note that any individual FDR threshold is quite arbitrary and has a clear meaning -- we should expect that at any FDR α , a fraction α of all significant metabolites are in fact false positives. This information can be incorporated into our inference equally well at an FDR of 0.1 or 0.05. With that said, we reran the analysis with the suggested cutoff of 0.05. The resulting analysis, as expected, resulted in a reduction of metabolites that were called significant, but lead to qualitatively identical conclusions on all major points reached in our manuscript. In the effort to make the analysis more transparent and more easily accessible to readers, we have annotated our analysis with an R markdown document. This document includes all computer code required to reproduce the analysis and the data were made available in the supplemental tables already. In addition, since we will make all our metabolomics data available (Supplementary Table 2), readers will be able to choose their FDR cutoff.

More generally, we note that we have applied fundamentally sound methods for analyzing the data in our study, and built all of our conclusions on the results obtained using those well-established methods. As long as the appropriate assumptions of the test are met, a particular p-value obtained using a hypothesis test is equally valid whether one has three, four, six, or ten biological replicates. As we have followed appropriate testing procedures, the only reasonable concern that could be raised regarding the number of replicates used is that we could have obtained better statistical power with more replicates. On that point we would certainly agree, but the massive expense (~ \$60,000) of additional replicates simply to slightly improve our detection of significant differences does not seem to us to be justified in light of the clear biological interpretation that we can already provide. As we detailed in the previous point, our procedures regarding biological replicates and data analysis in particular are indeed fully consistent with, or even exceed, the standards in the field of *Drosophila* metabolomics.

5. *"Flyscape is a good bioinformatic tool, but the general concept follows the previous publication (Metscape, ref 17). No major improvement was made to Flyscape. Only BioCyc*

database was used for Flyscape. How about the KEGG? How many of 391 metabolites are mapped to Flyscape? I think if the KEGG could be added to Flyscape, the coverage should be further improved.”

We thank the reviewer for the overall positive assessment of Flyscape. The overall functionality of Flyscape is similar to Metscape. Given the success of Metscape and the lack of similar tools for studying *D. melanogaster* metabolism we specifically chose to fashion Flyscape after Metscape. However, Flyscape is distinct from Metscape. The main power of Flyscape comes from providing appropriate biological context that can help interpret the data and generate new hypotheses. In addition to the information that has been incorporated into Flyscape database, the tool provides a number of links to various external resources including Flybase. This constitutes an important distinction between the two tools. As reviewer 1 stated, *“The development of Flyscape provides a powerful tool for Drosophila researchers. This application fills a major need in the community and will be widely used.”* As a proof of principle, we made Flyscape available on the Cytoscape app website in November. Without any active publicity, it has been downloaded ~200 times in just a few months.

We also share the reviewer’s views on the usefulness of KEGG, however KEGG data are no longer publicly available for download. Due to the lack of funding in 2011 the creators of KEGG were forced to go to a licensing model. It is our understanding that under this model the license holder is prohibited from redistributing the data. Including KEGG data into Metscape would constitute forbidden data redistribution. Further details can be found on the KEGG website (<https://www.genome.jp/kegg/docs/plea.html>).

6. *“The authors designed several feeding experiments to mimic different physiological/pathological conditions. But NO biological data was provided to characterize the flies (e.g., health status and behaviors). For example, after the 18-hour starvation, Drosophila usually underwent stress, and the autophagy may also be activated. The measured metabolic profiles indicated the response to stress (at least in part), instead of the effect of “refed”. Similar issues are with other designed experiments. Only experimental conditions are described, no biological data was provided to prove that the animal models are normal, and truly related to the aimed physiological/pathological conditions.”*

We agree with the reviewer that the inclusion of experiments that characterize the behaviors and physiology of the conditions tested would improve the manuscript. We took this suggestion and added two new main figures and several supplemental figures that address this point- this is detailed in the first part of the rebuttal letter. We believe that these behavioral assays and physiological measurements provide a context to analyze the metabolite changes uncovered by our work. Over the years many manuscripts have been published characterizing other aspects of the physiology of fasted and refed flies, such as circulating glucose levels and glycogen reserves³⁻⁶, or fasting times⁷⁻¹¹. The fasting time we used is in line with what the field uses and has been shown by many to result in the expected behavioral changes. We now describe in the text what our feeding manipulations have in common and how they differ from those others have used (and why we used them).

7. “Most conclusions in the manuscript were based on correlative analysis and literature reviewing. NO biological validated were provided. I hope the authors would validate at least one or two points through biological experiments.”

We have now added assays that measure triglyceride levels to show that fasting and diet alter their levels. However, we disagree with the reviewer 2 statement that our data is only based on correlative analysis and literature reviewing. As we explain in the introduction, our goal was to map the metabolites that change during the transition between hunger and satiety states in flies. While we know how the behaviors of hungry flies differs from that of sated animals, (incidentally, these differences are also found in rodents), we have no data on the metabolic changes that occur during this behavioral shift. For example, hungry flies suppress their sleep^{10,11}, are more tuned to food odors¹², have increased motivation to learn¹³, forage for food¹⁴, and choose nutritious over tastier, non-nutritive foods⁷⁻⁹. These behaviors change rapidly as animals become sated, a process called behavioral allostasis, but how this occurs is unknown. Studies in yeast and in mammalian tissues have shown that diet-derived metabolites can drive rapid changes in cell physiology¹⁵; if and how this impacts complex behavior is not known. Our goal was to map the changes in metabolic profiles between the hungry and sated state, and to ask how they are influenced by a high sugar diet, to create a starting point to address how they play a role in driving behavioral allostasis in fasted and sated flies. To this end, the identification of the metabolic signatures that characterize the heads of diet-induced obesity flies (Fig. 6), has already

served as a starting point to study the effects of a sugar diet, via increased activity of the hexosamine biosynthesis pathways, on taste perception in our lab (*May et al.*, in press). In conclusion, we think that our findings will move the field forward in several different ways. First, they will function as a molecular inroad to test the role of changing metabolites and their pathways in regulating responses to fasting and inducing satiety. Second, they will allow research to ask if and how the behavioral differences observed in animals fed a high sugar diet are controlled by changes in these metabolites. And finally, they will be useful datasets to those interested in metabolic disease and its effects on different tissues; this why we felt it was important to draw parallels and distinctions with the mammalian literature on diet-induced obesity. This was also one of the reasons to ensure that our data and manuscript are open access, since metabolomics data are usually not made publicly available in peer-reviewed publications.

Response to Reviewer 3 comments:

We thank the reviewer for their positive comments on our manuscript and their advice. We have taken the reviewer's suggestions, and focused our manuscript by adding feeding behavior measurements and limiting the number of meaningful comparisons drawn. We have addressed this point above in "Outline of major changes in revised manuscript."

*"Most experimental approaches in biology rely on recording the reaction of biological systems to experimental perturbations. This approach is especially relevant in "omics" approaches where a very large repertoire of parameters is recorded to measure the impact of experimental manipulations. These data are then used to reconstruct the rules or mechanisms governing the regulation of the interrogated biological system. Such approaches have been very powerful in many systems including *Drosophila melanogaster*. Given technological advances in instrumentation and bioinformatics, an approach gaining popularity is the comprehensive survey of metabolites in cells and organisms. This so-called metabolome is however highly complex and notoriously difficult to interpret. Nevertheless, the current rebirth of physiology and metabolic research and the realization that nutrition is key to our understanding of many diseases, makes the measurement and interpretation of metabolome information extremely relevant. In this context, it is curious that despite the track record of the *Drosophila* community in trailblazing the development and deployment of new methods, the tools available to the fly community to analyze and interpret metabolome information are very limited. Especially when it comes to integrating information from metabolomics and other "omics" approaches. Also, the number of metabolomics studies in the fly is (still) limited. This is the premise on which the work described in this manuscript is based. In the present manuscript, the authors describe a set of experiments in which they modify the availability of sugar in the diet of flies followed by metabolomics measurements and sometimes transcriptomics. They then use a*

newly developed tool to visualize their results in the context of largely already published transcriptomics data. The topic of the manuscript is therefore very timely and relevant for the Drosophila and metabolomics community. I have however some reservations about the overall structure of the manuscript and how some of the experiments have been designed and interpreted.”

1. “My main concern relates to how the work itself and the results are presented in the paper and how some analyses were performed. In general, I had the feeling that the authors often distort or exaggerate the impact of their study and what can be interpreted. For example, the title gives the impression that multiple dietary components are tested (sugars, vitamins, fatty acids, amino acids, nucleotides etc.) when in reality the authors only manipulate the sugar content of the diet. Also, the authors talk about feeding states in the title and it is not clear to me what they mean by that as they never quantify nor carefully look at feeding behavior in the animal. From performing a quick literature search it became apparent that some of the dietary manipulations used in the paper can lead to different changes in feeding behavior which seem to differ depending on the lab in which they are performed. This could be an additional concern and the authors should mention this and discuss the relevant papers. After reading the title and reading the abstract, reading the manuscript was a bit anticlimactic (and not because it is worse than suggested but because it is different to what was promised at the beginning).”

We are sorry that our manuscript came across this way, it is certainly not our intention to overstate the findings of our work. We have made changes to the title and throughout the text to moderate the language used and address this criticism. About feeding behaviors: we agree with the reviewer that the terminology we used in the original version of the manuscript was not clear. To this end, we have added behavioral experiments to characterize the feeding and foraging states of animals that were fasted or refed. We have addressed this point in full at the beginning of the letter (#2).

3. “I also found the design of the dietary manipulations a bit confusing. This then extends to the corresponding interpretation of the analysis of the metabolomics results. The authors perform a large set of experiments and then try to make generalizable interpretations. However, I must say that for me the different experiments are not easy to compare and do not help to generate a coherent picture. It is also not always clear to me why the authors first feed the fly, then starve it and then refeed it. I found that this complex design, which is augmented by the multitude of manipulations of the sugar content of the diet, leads to more confusion than clarity....The confusion is maybe augmented by a generally superficial and poor description of the content of

the paper at the start of the results section. I would find it helpful to have a more thorough explanation. It was not always clear for me what conditions were tested, what was the origin of the data, and for which purpose the experiments were performed.”

We apologize for not stating the rationale behind the design of our manipulations more clearly in the original version of the manuscript. We have addressed this by rewriting the introduction and the result sections describing Fig. 1 and Supplemental Figure 1, which characterize the feeding and foraging behaviors of fasted and refed flies. We chose to test fasted and refed flies because we were interested in identifying the metabolites that change during the switch between hunger and satiety. This is an interesting question, because we know that studies in yeast and mammalian cells find that influx of nutrients alters gene expression and drives physiological changes in cells and tissues. We want to know which metabolites change as a starting point to study if and how they play a role in the changes in neuronal excitability that underlie the shift between hunger and satiety. To this end, we refed flies with a concentration of D-glucose that rapidly leads to changes in the motivation to look for food (foraging behavior, Supplemental Figure 1) and in feeding behavior (Figure 1). We then used the same manipulations to collect the heads and bodies for metabolomics to identifying what metabolites change. It is true that many people, including my postdoctoral papers⁷⁻⁹, have used sated flies (instead of refed flies). Sated flies are flies that are taken out of a food vial or fasted for just a few hours. However, using sated flies would have not allowed us to map the metabolites that are changing rapidly upon refeeding and that may impact neural function.

We compared flies on a control and sugar diets because consumption of high-nutrient diets because these are known to change metabolites, cell physiology, and we and other have observed changed in behaviors that change with the internal state. Thus, our aim was to map shifts in metabolic profiles during hunger and satiety on a high sugar diet to see if and how these play a role in the behavioral changes observed. Our analysis in Figure 6 and Supplementary Figures 10 and 11 details the metabolic changes that occur with different exposures to a high sugar diet in fasted and refed fly heads; we identified the ~20 metabolites and a handful of pathways that define fasted and refed flies in the two diets. This gives us a molecular handle to address the function of each of these pathways in altering feeding behaviors on a high sugar diet. To this end, we have identified a role for one of these metabolites, glucosamine-6-phosphate, in blunting sweet taste sensation (*May et al, in press*). Thus, we know that the approach outlined in the

manuscript has been successful in identifying at least one metabolite that alters neuronal function and behavior. The observations in Fig. 6 will allow our lab and others to test their function in feeding and feeding-associated behaviors (such as sleep, learning and memory, and motivation). Thus, we are of the opinion that the identification of these metabolic pathways will be useful for both the *Drosophila melanogaster* neuroscience and metabolism communities. We hope this clarifies the rationale behind these manipulations and that the explanation in the text is sufficiently clear for the readers now.

4. *“An additional important problem I have with the design of the experiment is that the authors often rely on published transcriptomics data to compare the effects they see in their metabolomics measurements with possible changes in the transcriptome in the fly. First of all, it took me quite some time to find out which data were generated by the authors and which were extracted from databases. Second, metabolomics should be very sensitive to differences in diet and other experimental parameters. To which extent are the diets and the experimental manipulations in the present study and the studies in which the transcriptomics data were generated comparable? Given the provided evidence I have no way to judge this (the authors do not even provide the detailed composition of the medium for the animals). I am therefore skeptical to which extent the presented metabolomics data can be compared with the transcriptomics data.”*

We wrote in the Methods and Results sections which transcriptomics data were generated by us and which were from another group. The data on Figure 3 were generated by us and match the metabolomics and behavioral conditions with the exception that we used brains, instead of heads. We waited 1 extra hour after refeeding to collect the brains because we would expect gene expression changes to occur after refeeding- incidentally, this is identical to the time flies spend on the FLIC in Figure 1 (flies are in the FLIC for 1 h), so we know that the hunger and satiety changes are solidified by then.

The data used in Figure 4f are from Dobson et al. The diet given to flies in this study has similar concentrations of sugars (40% vs. 38%), but a different amount of protein (10 vs. ~15%). Thus, the reviewer is right that the conditions are not identical. We have clarified this difference in the manuscript text, but are open to removing this panel if the reviewer prefers.

All our experiments used the Bloomington Recipe, which is a standard recipe used by the Bloomington Drosophila Stock Center and many fly researchers. We have added the composition to the Methods.

5. *“Furthermore, I am not sure that the authors make a good case about the usefulness of FlyScape. When going through the manuscript I had the impression that most of the findings described by the authors could have been achieved by simply analyzing the metabolomics and the transcriptomics results separately. I am sure that FlyScape can be a powerful tool but I have some reservations: From what I can see given the description of the FlyScape tool it does not perform any statistical analysis of the data like pathway enrichment etc. but uses precomputed statistical values. It is, therefore, a pure visualization tool. While this is useful I am not sure that I am convinced by the examples provided by the authors.”*

We agree with the reviewer that we did not do Flyscape justice in the original version of the manuscript. To address these comments, we changed the figures to emphasize the functionality of Flyscape compared to other methods of analyzing and visualizing data. For example, in Figure 2 panels *a* and *c* we show the metabolomics data in differential display format and use Flyscape visualization in panels *b* and *d* to highlight difference in the way nutrients change in the heads and bodies between the refed/fasted condition. We also added a new panel in Figure 3 where we used Flyscape to merge the brain RNAseq and head metabolomics datasets to identify candidate enzymes that modulate this metabolic process in fasted flies. We did a similar analysis in Figure 4f. We also added a new supplemental figure to highlight the different visualization methods in Flyscape.

We agree with the reviewer that Flyscape will be a useful tool, something that was pointed out by reviewer 1 also, *“The development of Flyscape provides a powerful tool for Drosophila researchers. This application fills a major need in the community and will be widely used.”* Indeed, we already made Flyscape available in Cytoscape and in just a few months it has been downloaded ~200 times. One of the main challenges in analyzing metabolomics data from fruit flies is that there is no application that was built on *D. melanogaster* metabolic pathways. Thus, the main power of Flyscape comes from providing appropriate biological context that can help interpret the data and generate new hypotheses. Flyscape visualization is powerful because it allows researchers to understand changes in the context of pathways. We feel this is an

advantage to the multiple, pairwise displays in which metabolomics data are usually analyzed, as these do not allow a snapshot of how multiple compounds, enzymes, reactions, and even pathways are connected at the cellular level. In addition, Flyscape allows users to analyze multi-omic data simultaneously, which can be a powerful hypothesis generating tool to focus on particular candidates. Finally, because of the information we incorporated into the Flyscape database, this tool provides a number of links to various external resources including Flybase and KEGG, which help with the analysis.

Flyscape was explicitly designed as a pathway mapping and visualization tool. It does not perform statistical analysis, although it can incorporate and visualize the results of differential analysis (e.g. Student's t test or ANOVA) and/or the results of enrichment analysis performed with any of the existing and well-established tools (e.g. DAVID or GSEA for gene expression data, MSEA for metabolites etc.). Our decision not to include any statistical functions into Flyscape was motivated by several factors. First, we wanted to maintain a flexible modular structure for this tool. Instead of implementing a limited number of statistical methods we left the choice of most appropriate analysis techniques to the user, allowing them to visualize the results of their analysis in Flyscape. We also felt that while enrichment analysis techniques have been extremely useful for gene expression studies, the scope of their application for metabolomics has been limited mostly due to the insufficient coverage of the experimentally measured metabolites by existing pathway databases. Further, our experience with Metscape motivated us to create a similar tool for fruit fly community.

6. *“Finally, I am not convinced by all the specific findings the authors describe in the manuscripts. Especially when it comes to interpreting the meaning of the changes in specific metabolites. There are so many examples described and so many correlations made with papers (mainly from the vertebrate literature) that I am left to wonder how much I really learn from it. Often the overall findings feel trivial. Given that nothing is explored in depth and no follow-up experiments are performed my impression is that if you measure enough metabolites you will always find specific ones changing with your manipulation. And often these metabolites have been proposed to correlate with disease or condition x in a different system. Giving 1 or 2 examples would be enough. But only if the authors make a case of why this is relevant and how this can lead to important insights using the fly model. At this stage, it is just a long list of correlations, comparisons, and nothing more. In short, I think the paper is very relevant and timely but the paper tries to be too many things at the same time: a description of a new bioinformatics tool, an in-depth analysis of metabolomics findings, and an exhaustive description of different experimental manipulations. At the end, as a reader, I could not see the forest from the many trees and the resulting superficiality makes it very difficult for me to*

evaluate in depth what the authors have done and how well they have done it. I am convinced that if the authors focus on a limited set of messages, focus on a specific small set of manipulations and then dive deeply into them this will result in a clear and enriching manuscript. I general like the work and I am quite sure that FlyScape could be a useful tool but at this stage, I find it not suitable for publication in its current form.

We agree with the reviewer and think that the rewrite and the addition of the behavioral experiments give focus and highlight the rationale for our work: 1) to identify the metabolites that change during the fast transition between hunger and satiety in flies, and 2) to understand how these are influenced by consumption of a high sugar diet. Figures 2-3 now show with resolution we lacked before what compounds are changed in hungry and sated flies and how these differ in the heads and bodies. Figure 4-6 focus instead on diet, highlighting its effect on body metabolism and feeding behaviors (Fig. 4 and 5) and identifying the metabolic signatures of the heads of fasted and sated flies on a sugar diet (Fig. 6). We expanded on this point in above where the major changes are outlined.

We disagree that our findings are trivial: rather, we expect that they will move the field forward in several different ways. First, they will function as a molecular inroad to test the role of these metabolites and their pathways in regulating responses to fasting and inducing satiety. Second, they will allow research to ask if and how the behavioral differences observed in animals fed a high sugar diet are controlled by changes in these metabolites. And finally, they will be useful dataset to those interested in metabolic disease and its effects on different tissues.

Minor comments:

1. *“I would plot the number of metabolites changing in a specific class as % and not the total number. It could be that for example many lipids are found changing because the measurement method is really good at detecting lipids.”*

We have changed the way the changes are reported in the figures.

2. *“I would recommend that the authors try to avoid hyperbole like “unparalleled resource” (line 76).”*

We have removed this.

3. *“Given that glia play a key role in the metabolic dynamics of the brain the authors should give that interpretation of their results and the related literature (especially in Drosophila) more attention. Especially for the finding described in lines 190-92.”*

We addressed this in the discussion in the original manuscript. We feel the discussion is a better place to address this, but are happy to add the discussion here too.

4. *“The title in line 207 is highly trivial.”*

We have changed this heading.

5. *“There are so many comparisons in the paper that I often lost track what the authors are comparing to what (line 219 for example).”*

We have cut back on the number of relevant comparisons.

6. *“Feeding state and nutrient state are used a lot and it is not clear what the authors mean by this. They should avoid this term or clarify it.”*

We have changed these and simply refer to fasted/refed or hungry/sated flies.

7. *“The numbers of metabolites in the text and the figure sometimes do not match (Figure 1).”*

Fixed.

8. *“The statements in line 421-423 and the end of the discussion are overreaching quite a bit. The experiments on their own are interesting enough and the physiology of animals fascinating enough. Except if the authors can be specific those parts are just marketing exaggerations.”*

Fixed.

9. *“The authors should include the detailed composition of the so-called “Bloomington recipe” and also describe carefully how the transcriptomics experiments they used from the literature were done and how those conditions compare to the metabolomics ones.”*

The reviewer made the same point in #4 above, we responded there.

10. “The colors used in the figures are often very similar and it is difficult to differentiate the different metabolite classes in the plots.”

We used the color wheel from paletton.com to reassign the compound color classes to make them as contrasting as possible.

References

- 1 Ro, J., Harvanek, Z. M. & Pletcher, S. D. FLIC: high-throughput, continuous analysis of feeding behaviors in *Drosophila*. *PLoS One* **9**, e101107, doi:10.1371/journal.pone.0101107 (2014).
- 2 Bratty, M. A., Chintapalli, V. R., Dow, J. A., Zhang, T. & Watson, D. G. Metabolomic profiling reveals that *Drosophila melanogaster* larvae with the y mutation have altered lysine metabolism. *FEBS Open Bio* **2**, 217-221, doi:10.1016/j.fob.2012.07.007 (2012).
- 3 Chintapalli, V. R., Al Bratty, M., Korzekwa, D., Watson, D. G. & Dow, J. A. Mapping an atlas of tissue-specific *Drosophila melanogaster* metabolomes by high resolution mass spectrometry. *PLoS One* **8**, e78066, doi:10.1371/journal.pone.0078066 (2013).
- 4 Tennessen, J. M. *et al.* Coordinated metabolic transitions during *Drosophila* embryogenesis and the onset of aerobic glycolysis. *G3 (Bethesda)* **4**, 839-850, doi:10.1534/g3.114.010652 (2014).
- 5 <https://www.metabolomicsworkbench.org/>
- 6 Paley, S. M. & Karp, P. D. The Pathway Tools cellular overview diagram and Omics Viewer. *Nucleic acids research* **34**, 3771-3778, doi:10.1093/nar/gkl334 (2006).
- 7 Dus, M., Ai, M. & Suh, G. S. Taste-independent nutrient selection is mediated by a brain-specific Na⁺ /solute co-transporter in *Drosophila*. *Nat Neurosci* **16**, 526-528, doi:10.1038/nn.3372 (2013).
- 8 Dus, M. *et al.* Nutrient Sensor in the Brain Directs the Action of the Brain-Gut Axis in *Drosophila*. *Neuron* **87**, 139-151, doi:10.1016/j.neuron.2015.05.032 (2015).
- 9 Dus, M., Min, S., Keene, A. C., Lee, G. Y. & Suh, G. S. Taste-independent detection of the caloric content of sugar in *Drosophila*. *Proc Natl Acad Sci U S A* **108**, 11644-11649, doi:10.1073/pnas.1017096108 (2011).
- 10 Keene, A. C. *et al.* Clock and cycle limit starvation-induced sleep loss in *Drosophila*. *Curr Biol* **20**, 1209-1215, doi:10.1016/j.cub.2010.05.029 (2010).
- 11 Lee, G. & Park, J. H. Hemolymph sugar homeostasis and starvation-induced hyperactivity affected by genetic manipulations of the adipokinetic hormone-encoding gene in *Drosophila melanogaster*. *Genetics* **167**, 311-323 (2004).
- 12 Root, C. M., Ko, K. I., Jafari, A. & Wang, J. W. Presynaptic facilitation by neuropeptide signaling mediates odor-driven food search. *Cell* **145**, 133-144, doi:10.1016/j.cell.2011.02.008 (2011).
- 13 Burke, C. J. & Waddell, S. Remembering nutrient quality of sugar in *Drosophila*. *Curr Biol* **21**, 746-750, doi:10.1016/j.cub.2011.03.032 (2011).
- 14 Hergarden, A. C., Tayler, T. D. & Anderson, D. J. Allatostatin-A neurons inhibit feeding behavior in adult *Drosophila*. *Proc Natl Acad Sci U S A* **109**, 3967-3972, doi:10.1073/pnas.1200778109 (2012).

- 15 Lu, C. & Thompson, C. B. Metabolic regulation of epigenetics. *Cell metabolism* **16**, 9-17, doi:10.1016/j.cmet.2012.06.001 (2012).

Reviewers' comments:

Reviewer #1 (Remarks to the Author):

The authors have addressed my concerns.

Reviewer #2 (Remarks to the Author):

First of all, I thank the authors' effort to thoroughly revise the manuscript. However, most of my major concerns still exist after the revision (Reviewer 2).

In general, I agree with Reviewer 3 for the following two points "the authors often distort or exaggerate the impact of their study and what can be interpreted" and "the paper tries to be too many things at the same time: a description of a new bioinformatics tool, an in-depth analysis of metabolomics findings, and an exhaustive description of different experimental manipulations". After the extensive revision, the two major concerns still exist.

My personal suggestion is to organize the Flyscape part as a separated bioinformatic paper and submit to a specialized journal such as Bioinformatics or BMC Bioinformatics as a software tool. Then, the authors could focus on biological findings and Drosophila metabolomics in this work.

Second, I like the idea that the authors try to conclude in the revised manuscript, "understanding the role of metabolites in modulating complex behavior". However, again, the authors did not provide any biological validation results. I wish to see a genetic manipulation experiment (by knock-out, knock-in, or mutation) that proves the regulation of the specific metabolite levels (i.e., those found in the metabolomics experiments) in fly could rescue the fly behaviors. Instead, a metabolite feeding experiment is also applicable for validation. The authors must establish a causal relationship instead of descriptive report.

Other specific comments to the authors' responses:

Response to Comment 1: the cited papers by the authors were published during 2012-2014. Now in 2019, the metabolomics technology improved a lot. Even Metabolon routinely measures >500 metabolites per sample in their service. In addition, many dysregulated metabolites are lipids. It is the reason that I asked for a lipidomics analysis.

Response to Comment 2&3: data quality is still the major concern. In the revision, the authors mentioned the RSD for the quality control. Page 38, Line 801-802: "The Relative Standard Deviation (RSD) for each compound measured in the pooled samples was calculated and the median reported in the text". In metabolomics, any compound in QC samples with RSD>30% should be removed from the subsequent analysis. The medium RSD is not enough for data quality evaluation. Please provide the raw data for "each compound measured in the pooled samples", and list the calculated RSD for each compound.

In addition, the data table has too many missing values (MV, i.e., NA). I had a rough counting, and found that 86 out of 391 metabolites have more than 50% missing values. In metabolomics, metabolites with MV>50% should be removed from the subsequent analysis.

The number of biological replicates in this study is not enough. I cannot agree with the author's argument. The authors can consult a statistician, and calculate the minimal required replicate number (or statistical power) with the current measured biological variability in their metabolomics data set. Please provide the calculation data.

Response to Comment 4: I cannot agree to use p-value cut-off as 0.1. For example, in RNA-seq data, the authors used 0.05 as p-value cut-off. The authors should report the analysis result with p-value cut-off of 0.05. In the revised manuscript, the authors still used the 0.1 as the p-value cut-off.

Response to Comment 7: the concern still exist, how metabolites influence behavior is unclear, and should be addressed through a mechanistic validation experiment.

Reviewer #3 (Remarks to the Author):

The revision has clearly improved the Wilinski et al. manuscript. The reorganization of the figures and the focused narrative has greatly improved the flow of the paper and helped clarify some key questions remaining after reading the original manuscript. Also, the addition of behavioral data

clearly strengthens the manuscript. Some easy to resolvable issues remain which I will mention below. A key issue of the paper remains that it is largely descriptive and that the authors do not use a key advantage of their experimental system (*Drosophila m.*) allowing them to mechanistically test findings or predictions from their data. The reference to an unpublished paper (May et al.) is not helpful as I cannot assess it. It is not even available as a preprint. Also, the authors claim to have incorporated suggestions from my previous review but failed to do so. This is disappointing.

This said the paper still contains valuable information and represents as much as I could see one of the most exhaustive metabolomics analysis in this model organism. As such it contains valuable information worth publishing in a good journal. I therefore support the publication of a revised version in Nature Communications.

Major concerns:

I still think that the paper has some flaws in terms of overstating the findings. The new title is much better but I strongly suggest the authors change “diet-dependent” by “sugar-dependent” or “carbohydrate-dependent” in the title. That is what they do and diet is too loose a term to be used there. Also, in the abstract, the authors should not state that they identified a “key metabolic transition”. How do they know it is a key transition? They never test it. Again, in the abstract, the link to complex behaviors is clearly overstated. At this stage, it is not clear in any way that the changes in metabolic landscape mapped by the authors have any link to “complex behaviors”. Currently it reads like there might be a link between the “key metabolic transition” and the changes in behavior reported. This is never shown. The authors should therefore not make that link in the abstract.

In general, the authors should downplay the relationship between their study and understanding complex behaviors. At this stage, there is no proof that there is a link between the observed metabolic changes and the behavioral changes. Changes in behavior could, for example, be mediated by interorgan communication (as often happens in vertebrates, see leptin). The authors should discuss this possibility in the discussion. They should at least mention that there might be no link between changes in metabolites and changes in behavior and mention interorgan communication as a possibility.

In my previous comment, I had suggested that the authors discuss the contribution of glia to energy usage in the brain and how that could explain some differences observed between heads and the body and corresponding literature. I found, for example, a nice paper showing this in *Drosophila* (Volkenhoff et al., Cell metabolism 2015). Despite their claim in the rebuttal, the authors do not do so in the current manuscript. Glia metabolize sugars and provide neurons with energy in the form of lactate and pyruvate. This fits very nicely with the data of the authors. This should be included and discussed more extensively than with just one reference!

The behavioral data added by the authors clearly strengthen the paper. I have however some concerns regarding their interpretation and statistical analysis.

- 1) The authors do not mention which type of statistical analysis was used when analyzing the data in figure 1. This is key in assessing their interpretation.
- 2) The data look like they are not normally distributed. The authors nevertheless use means and other summary statistics which are best used when analyzing normally distributed data. I would suggest the use of medians and CIs. Given that the data are likely not to be normally distributed the type of statistical test used in the analysis should also be chosen accordingly.
- 3) The authors write in the main text that they define a specific parameter in the feeding behavior as “licks”. Looking at the description of the method used by the authors and in analogy to licking in vertebrates I am doubtful that the authors are truly measuring a motor output which is related to licks. For this, the authors would need to sample the behavior at a much higher rate. Also, the authors never look directly at the feeding behavior of the animal but measure it indirectly. They have no idea what they are measuring except for the interaction of the animal with food. The authors should therefore not use the term lick but a more neutral term.

I find the observation that neurotransmitters change in the head dataset trivial. That tissue contains mainly the brain and therefore that has to be the tissue where these chemicals change. I would suggest the authors downplay their claims there.

While the correlation between some metabolites and the behavior mentioned on page 19 are neat I wonder if this could not just be an artifact of sampling many metabolites. If you test enough you will end up finding a few ones which are correlated. Especially if they were filtered in advance to show a regulation by diet and if diet correlates with behavior. Ideally, the authors would test this finding themselves using a different method as you would do for genes using qPCR. If they do not experimentally validate this correlation the authors should at least do a multiple comparison correction or as an absolute minimum mention this limitation in the text and the importance of following this observation up to test if it holds up using different tests.

Minor concerns:

The authors should state in the main text that they used males. This is relevant when discussing the experimental design as males and females have very different kinetics in their response to starvation.

The authors claim that CG1673 has never been FUNCTIONALLY linked to a metabolic pathway and imply in line 271 that they do so. This study, however, cannot make any functional claims either as it is purely correlative. I would suggest the authors remove the indication that they do so. At this stage, their observations are largely related to the annotation of the function of that gene in FlyBase and do not go beyond that. Their findings are just compatible with the annotation and warrant further functional analysis of this gene.

Line 336-337 the authors claim that they are assessing the effect of dietary changes on the internal state of the animal by looking at their behavior. This is however not correct. There might be changes in the internal state which are not revealed by changes in behavior and changes in behavior do not have to be related to changes in internal state. This should be rephrased.

I can see the rationale behind collecting the tissue for RNAseq one hour later than the tissue for metabolomics but I was still surprised when I read this in the materials and methods. For transparency sake, this should be mentioned in the main text.

MONICA DUS, Assistant Professor

June 17, 2019

We thank the reviewers for their thoughtful comments and advice. We have addressed each reviewer's comments individually below in blue.

Reviewer #1 (Remarks to the Author):

The authors have addressed my concerns.

Reviewer #2 (Remarks to the Author):

First of all, I thank the authors' effort to thoroughly revise the manuscript. However, most of my major concerns still exist after the revision (Reviewer 2).

In general, I agree with Reviewer 3 for the following two points "the authors often distort or exaggerate the impact of their study and what can be interpreted" and "the paper tries to be too many things at the same time: a description of a new bioinformatics tool, an in-depth analysis of metabolomics findings, and an exhaustive description of different experimental manipulations". After the extensive revision, the two major concerns still exist.

My personal suggestion is to organize the Flyscape part as a separated bioinformatic paper and submit to a specialized journal such as Bioinformatics or BMC Bioinformatics as a software tool. Then, the authors could focus on biological findings and Drosophila metabolomics in this work.

It was certainly not our intent to exaggerate the impact of our findings, and at the suggestion of reviewer 3 we made considerable changes to the text. We have also added additional changes to the writing to address this point in the current revision #2 version. Indeed, it was our original intention to prepare two separate manuscripts, one for Flyscape and one for the Dm metabolomics data, and even contacted the editors at BMC and had editorial interest for a Flyscape manuscript. However, we reasoned that a manuscript that described both together would be a better resource for the community. Since metabolomics is becoming more mainstream in the fruit fly (and in other models), and is often used by labs in the fields of developmental biology, neuroscience, and development, we thought that reporting this tool together with that data in a non-specialist journal would increase the usefulness of the tool by making it more visible. Indeed, when considering the journals for submission, we explicitly looked for those which were open access and multidisciplinary. We settled on *Nature Communications* because this journal has a history of publishing resource articles that open biological questions, and we were inspired by Marshall OJ and Brand A, 2017. We are of course planning to follow up on the role of the metabolites identified in feeding behavior, but in such a manuscript there would also be no space to represent and analyze the data to the extent we have done in this manuscript. We believe that one of the important aspects of this work is that it will be useful to different scientists asking diverse questions. Those of us interested in feeding behavior can focus on the effect of the sugar diet on the metabolic profiles of heads, those interested in motivated behaviors can focus on the control diet head data, and those interested in obesity, metabolic disease and cancer, will use the body control and sugar diet metabolomics data.

Second, I like the idea that the authors try to conclude in the revised manuscript, “understanding the role of metabolites in modulating complex behavior”. However, again, the authors did not provide any biological validation results. I wish to see a genetic manipulation experiment (by knock-out, knock-in, or mutation) that proves the regulation of the specific metabolite levels (i.e., those found in the metabolomics experiments) in fly could rescue the fly behaviors. Instead, a metabolite feeding experiment is also applicable for validation. The authors must establish a causal relationship instead of descriptive report.

Our end goal is to understand the role of metabolites in complex behavior, but as we are sure the reviewer appreciates, that is easily a 10+ year question. We specifically stated in the introduction and the manuscript that the goal of this manuscript was to identify the metabolites that change during the transition between hunger and satiety, since this transition is associated with different behaviors, from sleep, to learning and memory, foraging, locomotion, and feeding. We did not claim in the manuscript to have established a causal connection between the two, but to have provided the first step to now begin asking that question. We have further modified the text at the suggestion to downplay our findings.

Other specific comments to the authors’ responses:

Response to Comment 1: the cited papers by the authors were published during 2012-2014. Now in 2019, the metabolomics technology improved a lot. Even Metabolon routinely measures >500 metabolites per sample in their service. In addition, many dysregulated metabolites are lipids. It is the reason that I asked for a lipidomics analysis.

The samples used for this work were collected in 2016 shortly after one of the corresponding authors opened her lab. JMK, the scientist at Metabolon who did the experiments on our samples and is an author on the manuscript, confirmed that 391 named biochemicals is typical of a 2015/early 2016 fruit fly study, where we wouldn’t see the biochemical diversity that we would in human or mouse samples, being a “simpler” model with limited range of dietary intake. In the current Metabolon platform architecture, 400-500 named metabolites could be expected, but that this would be still strongly influenced by the continued addition of biochemicals into the Metabolon compound library. The development of Flyscape started after funding was secured (U24 DK097153) and people with the right skillset were hired, as well as the hiring of DW to analyze the data and work with the AK lab to develop Flyscape.

Response to Comment 2&3: data quality is still the major concern. In the revision, the authors mentioned the RSD for the quality control. Page 38, Line 801-802: “The Relative Standard Deviation (RSD) for each compound measured in the pooled samples was calculated and the median reported in the text”. In metabolomics, any compound in QC samples with RSD>30% should be removed from the subsequent analysis. The medium RSD is not enough for data quality evaluation. Please provide the raw data for “each compound measured in the pooled samples”, and list the calculated RSD for each compound. In addition, the data table has too many missing values (MV, i.e., NA). I had a rough counting, and found that 86 out of 391 metabolites have more than 50% missing values. In metabolomics, metabolites with MV>50% should be removed from the subsequent analysis. The number of biological replicates in this study is not enough. I cannot agree with the author’s argument. The authors can consult a statistician, and calculate the minimal required replicate number (or statistical power) with the current measured biological viability in their metabolomics data set. Please provide the calculation data.

Missing values: As we described in the methods, compounds with 50% or more missing values were removed from analysis. This criteria was applied to each the pairwise comparisons (page 5, “mSet<-RemoveMissingPercent(mSet, percent=0.5)”) of R markdown (Supplementary Text 2), which is an important point because many of the metabolites we measure are only present in a particular tissue or condition. NAA is a clear example of this. NAA is a brain specific metabolite, in fact it is the second most abundant metabolite in human brains, so NAA is present in the head samples, but absent from the body

samples. If we followed the reviewer's request, we would remove NAA, because it has missing values in bodies. However, that would lead to the deletion of important metabolites just because of the heterogeneity of the samples. Thus, when we removed the samples for analysis we compared them to the same body part and condition.

Biological replicates: we disagree with the reviewer on this point. In addition to the references we provided in the rebuttal letter to show that our sample sizes are in line with what has been largely used in *Drosophila melanogaster* articles, we looked at additional recent articles published by the Thummel and Reed labs, which are two of the leading labs in fruit fly metabolomics. In Storelli et al, *Developmental Cell*, 2019, Thummel and colleagues use 4-5 replicates, and in Chialvo et al, *Metabolomics*, 2016, Reed and colleagues used 3 replicates. Further, another recent article in *Drosophila melanogaster* published by David Raftery's lab, used 3 replicates of 36 larvae each. Of note, in each of these studies, there was no mention of the RSD, CV or IQR cutoffs.

RSD: We have added the RSD values to the compounds reported in Supplementary Table 1, which gives readers the opportunity to draw their own conclusions about the data. In our data the technical replicate median RSD was around 3 and 7 % and the median RSD 30%. Metabolon's compounds are proprietary and as such, the company does not release the data on the RSD of each compound of the technical replicate samples. However, in the published study where the authors included technical replicates in their Metabolon samples and analysis, the "median intraclass correlation coefficient, calculated based on the quality-control samples, was 0.90 with an IQR of 0.74–0.96, suggesting a very high reproducibility" (Wang et al. *J Nutr.* 2018 Jun 1;148(6):932-943). With respect to the biological variation and the reviewer's suggestion to exclude biochemicals with an RSD >30%, we feel that it would be a mistake to exclude the compounds solely based on an arbitrary RSD cutoff. Given that our samples were collected from complex tissues from animals undergoing feeding perturbations, one would expect to see a higher level of variation compared to samples from cell culture. However, the statistical tests employed in our study are designed to select against compounds with very high noise and subtle differences. Clearly, RSD is an important benchmark in metabolomics studies, but it is not applied arbitrarily in the broad field of metabolism. For example, in the last dozen metabolomics papers published in *Nature Communications* only two reported RSDs and applied a cutoff. The application of a strict cutoff for biological variation obviously makes sense if the study is trying to identify a critical biomarker for diagnostic purposes using human plasma or urine, but not in the case of animal studies aimed at capturing global changes that correlate with a complex behavior or phenomenon. Further, in our study samples from each condition clearly separate from one another with high confidence intervals, showing that the biological variation is meaningful and not just noise.

Power analysis: We also carried out a retrospective power analysis on our metabolomic data, focusing on the ability to detect biologically meaningful changes at $p=0.05$, given the signal:noise ratios observed for compounds in our data set. As we show, for the most common case in our data (five replicates, $SD=0.4$ in \log_2 space) our retrospective power to detect a two-fold change in metabolite level is ~ 0.64 . Thus, while we cannot say with confidence that all changes of interest would be detected in our experiments, we achieve substantial power in identifying metabolites with two-fold or greater change. As with all statistical aspects of our data set, we have made all aspects of our methods completely available and transparent, so that readers may reach their own conclusions regarding our findings.

Response to Comment 4: I cannot agree to use p-value cut-off as 0.1. For example, in RNA-seq data, the authors used 0.05 as p-value cut-off. The authors should report the analysis result with p-value cut-off of 0.05. In the revised manuscript, the authors still used the 0.1 as the p-value cut-off.

The data table lists the p value and q value for each compound, so the readers will be able to analyze the data with their own cutoffs. To aid this, we have used colors in Supplementary Data 1 (old supp table 1) to differentiate compounds >0.05 (light blue) with those >0.1 (light green) FDR.

Response to Comment 7: the concern still exist, how metabolites influence behavior is unclear, and should be addressed through a mechanistic validation experiment.

We have addressed this point above; as we clearly state in the abstract and introduction, the scope of this work and this manuscript is to set a framework to study this question by identifying the metabolites that change during the transition between hunger and satiety. We are following up on these metabolites, but any study about a particular metabolite will require experiments and genetic manipulations, that as the reviewer surely appreciate, are bound to take a long time and create a large body of work that will be better presented in new publications.

Reviewer #3 (Remarks to the Author):

The revision has clearly improved the Wilinski et al. manuscript. The reorganization of the figures and the focused narrative has greatly improved the flow of the paper and helped clarify some key questions remaining after reading the original manuscript. Also, the addition of behavioral data clearly strengthens the manuscript. Some easy to resolvable issues remain which I will mention below. A key issue of the paper remains that it is largely descriptive and that the authors do not use a key advantage of their experimental system (*Drosophila m.*) allowing them to mechanistically test findings or predictions from their data. The reference to an unpublished paper (May et al.) is not helpful as I cannot assess it. It is not even available as a preprint. Also, the authors claim to have incorporated suggestions from my previous review but failed to do so. This is disappointing.

This said the paper still contains valuable information and represents as much as I could see one of the most exhaustive metabolomics analysis in this model organism. As such it contains valuable information worth publishing in a good journal. I therefore support the publication of a revised version in Nature Communications.

We thank the reviewer for their comments and details on which part of the manuscript to address. The manuscript is now in press at May et al, Cell Reports, 2019.

Major concerns:

I still think that the paper has some flaws in terms of overstating the findings. The new title is much better but I strongly suggest the authors change “diet-dependent” by “sugar-dependent” or “carbohydrate-dependent” in the title. That is what they do and diet is too loose a term to be used there. Also, in the abstract, the authors should not state that they identified a “key metabolic transition”. How do they know it is a key transition? They never test it. Again, in the abstract, the link to complex behaviors is clearly overstated. At this stage, it is not clear in any way that the changes in metabolic landscape mapped by the authors have any link to “complex behaviors”. Currently it reads like there might be a link between the “key metabolic transition” and the changes in behavior reported. This is never shown. The authors should therefore not make that link in the abstract.

In general, the authors should downplay the relationship between their study and understanding complex behaviors. At this stage, there is no proof that there is a link between the observed metabolic changes and the behavioral changes. Changes in behavior could, for example, be mediated by interorgan communication (as often happens in vertebrates, see leptin). The authors should discuss this possibility in the discussion. They should at least mention that there might be no link between changes in metabolites and changes in behavior and mention interorgan communication as a possibility.

We thank the reviewer for detailing these issues and we have addressed them by editing the writing in the manuscript.

In my previous comment, I had suggested that the authors discuss the contribution of glia to energy usage in the brain and how that could explain some differences observed between heads and the body and corresponding literature. I found, for example, a nice paper showing this in *Drosophila* (Volkenhoff et al.,

Cell metabolism 2015). Despite their claim in the rebuttal, the authors do not do so in the current manuscript. Glia metabolize sugars and provide neurons with energy in the form of lactate and pyruvate. This fits very nicely with the data of the authors. This should be included and discussed more extensively than with just one reference!

We have extended this point in the discussion (starting at line 497).

The behavioral data added by the authors clearly strengthen the paper. I have however some concerns regarding their interpretation and statistical analysis.

1) The authors do not mention which type of statistical analysis was used when analyzing the data in figure 1. This is key in assessing their interpretation.

We now provide a detailed description of the approaches used in analyzing the experimental data -- as noted below, our new approaches also directly address the reviewer's specific concerns on the appropriateness of statistical methods.

2) The data look like they are not normally distributed. The authors nevertheless use means and other summary statistics which are best used when analyzing normally distributed data. I would suggest the use of medians and CIs. Given that the data are likely not to be normally distributed the type of statistical test used in the analysis should also be chosen accordingly.

We have re-analyzed these data using specific methods for each experimental dataset. For count data (eg, licks and feeding event numbers), we now use a zero-inflated negative binomial model to treat the data, which provides an excellent fit for the actual observed distributions. For other quantitative data on feeding, we now report medians and confidence intervals (the latter calculated using stratified bootstraps), as the form of the underlying distribution is unclear.

3) The authors write in the main text that they define a specific parameter in the feeding behavior as "licks". Looking at the description of the method used by the authors and in analogy to licking in vertebrates I am doubtful that the authors are truly measuring a motor output which is related to licks. For this, the authors would need to sample the behavior at a much higher rate. Also, the authors never look directly at the feeding behavior of the animal but measure it indirectly. They have no idea what they are measuring except for the interaction of the animal with food. The authors should therefore not use the term lick but a more neutral term.

We used the term 'lick' because this is what was used in the first manuscript (and many subsequent ones) characterizing the FLIC, but wrote in the results section that we are measuring "real-time interactions of flies with the food at 5Hz". A longer explanation of the FLIC is also present in the methods. We changed the word "licks" to "feeding interaction" in the text and figures.

I find the observation that neurotransmitters change in the head dataset trivial. That tissue contains mainly the brain and therefore that has to be the tissue where these chemicals change. I would suggest the authors downplay their claims there.

Most neurotransmitters are derived from amino acid metabolism, which is also present in bodies (for example, most of the body serotonin is found in the bowels, not in brains in humans, while free glutamate is the most abundant metabolite in the human brain), but we observed changes in the precursors and of these compounds only in heads. These data are presented as a supplementary figure, so it is not a central point of the manuscript already, but we have edited the writing to downplay it further.

While the correlation between some metabolites and the behavior mentioned on page 19 are neat I wonder if this could not just be an artifact of sampling many metabolites. If you test enough you will end up finding a few ones which are correlated. Especially if they were filtered in advance to show a regulation

by diet and if diet correlates with behavior. Ideally, the authors would test this finding themselves using a different method as you would do for genes using qPCR. If they do not experimentally validate this correlation the authors should at least do a multiple comparison correction or as an absolute minimum mention this limitation in the text and the importance of following this observation up to test if it holds up using different tests.

It was an oversight on our part not to include the p value for the metabolites found through the Pearson correlation. While the metabolites we reported in Figure 6 had a significant p-value, when we calculated their FDRs within the context of the whole set of metabolites, we did find a FDR rate of 0.38 for the most significant 29 hits. This suggests that on average if we would screen all the metabolites in this list (many of which are in similar pathways) with genetics manipulations, we would expect that 40% would be false positive (as long as our statistical assumptions hold). As far as screens go, we believe that a hit rate of ~50% would represent a promising first tier of analysis for identifying compounds of potential interest, but given that we have not tested the effects of any of these genetically with the exception of glucosamine-6-phosphate (May et al, Cell Reports, 2019), we wouldn't want to hang too strongly on them and so decided to remove the graphs from figure 6. That said, we think it would be beneficial to list the correlations on a supplemental table.

Minor concerns:

The authors should state in the main text that they used males. This is relevant when discussing the experimental design as males and females have very different kinetics in their response to starvation.

We have added an explanation as to why we used males in the methods session. While it is true that males and females have different starvation kinetics, flies of both sexes will suppress sleep, and show motivated behaviors such as food seeking, odor tracking, and nutrient sensing when fasted for ~20 hours. Female will survive starvation longer compared to males (1.5-2 vs 2-2.5 days). Males and mated females also have differential preference for nutrients.

The authors claim that CG1673 has never been FUNCTIONALLY linked to a metabolic pathway and imply in line 271 that they do so. This study, however, cannot make any functional claims either as it is purely correlative. I would suggest the authors remove the indication that they do so. At this stage, their observations are largely related to the annotation of the function of that gene in FlyBase and do not go beyond that. Their findings are just compatible with the annotation and warrant further functional analysis of this gene.

We agree with the reviewer that our findings do not provide any evidence for the function of the gene, but disagree that line 271 implies that we do so, however, we appreciate that the writing could be made clearer and have amended it.

Line 336-337 the authors claim that they are assessing the effect of dietary changes on the internal state of the animal by looking at their behavior. This is however not correct. There might be changes in the internal state which are not revealed by changes in behavior and changes in behavior do not have to be related to changes in internal state. This should be rephrased.

We have rephrased this.

I can see the rationale behind collecting the tissue for RNAseq one hour later than the tissue for metabolomics but I was still surprised when I read this in the materials and methods. For transparency sake, this should be mentioned in the main text.

We have added this to the main text.

Fig. 1 Retrospective power analysis of metabolomics data.

- a)** Histogram of observed variation (standard deviation) in metabolic compounds from stated and starved flies across conditions in bodies (CD, SD) and heads (CD, SD2, SD5, and SD7).
- b)** Plot of expected power for 3, low (Q1), medium (median), and high (Q3), levels of variation using $n = 5$ and a significance level of 0.05.

PowerAnalysis

Daniel Wilinski

load data

```
data <- read.csv("metabolonOriginalData.csv", header = TRUE, row.names = 5)

ND_bodies <- c("a7D_ND_sated_bodies_1", "a7D_ND_sated_bodies_2", "a7D_ND_sated_bodies_3",
  "a7D_ND_sated_bodies_4", "a7D_ND_sated_bodies_5", "a7D_ND_starved_bodies_6",
  "a7D_ND_starved_bodies_7", "a7D_ND_starved_bodies_8", "a7D_ND_starved_bodies_9",
  5, 4)
ND_heads <- c("a7D_ND_sated_heads_6", "a7D_ND_sated_heads_7", "a7D_ND_sated_heads_8",
  "a7D_ND_sated_heads_9", "a7D_ND_sated_heads_10", "a7D_ND_starved_heads_1",
  "a7D_ND_starved_heads_2", "a7D_ND_starved_heads_3", 5, 3)
SD7_bodies <- c("a7D_HSD_sated_bodies_21", "a7D_HSD_sated_bodies_22", "a7D_HSD_sated_bodies_23",
  "a7D_HSD_sated_bodies_24", "a7D_HSD_sated_bodies_25", "a7D_HSD_starved_bodies_26",
  "a7D_HSD_starved_bodies_27", "a7D_HSD_starved_bodies_28", "a7D_HSD_starved_bodies_29",
  "a7D_HSD_starved_bodies_30", 5, 5)
SD2_heads <- c("a2D_HSD_sated_heads_41", "a2D_HSD_sated_heads_42", "a2D_HSD_sated_heads_43",
  "a2D_HSD_sated_heads_44", "a2D_HSD_sated_heads_45", "a2D_HSD_starved_heads_46",
  "a2D_HSD_starved_heads_47", "a2D_HSD_starved_heads_48", "a2D_HSD_starved_heads_49",
  "a2D_HSD_starved_heads_50", 5, 5)
SD5_heads <- c("a5D_HSD_sated_heads_32", "a5D_HSD_sated_heads_33", "a5D_HSD_sated_heads_34",
  "a5D_HSD_sated_heads_35", "a5D_HSD_starved_heads_36", "a5D_HSD_starved_heads_37",
  "a5D_HSD_starved_heads_38", "a5D_HSD_starved_heads_39", 4, 4)
SD7_heads <- c("a7D_HSD_sated_heads_11", "a7D_HSD_sated_heads_12", "a7D_HSD_sated_heads_13",
  "a7D_HSD_sated_heads_14", "a7D_HSD_sated_heads_15", "a7D_HSD_starved_heads_16",
  "a7D_HSD_starved_heads_17", "a7D_HSD_starved_heads_18", "a7D_HSD_starved_heads_19",
  "a7D_HSD_starved_heads_20", 5, 5)

listofvectors <- list(ND_bodies, ND_heads, SD7_bodies, SD2_heads, SD5_heads,
  SD7_heads)

for(dataname in listofvectors) {
  data_subset <- data %>%
    select(dataname[1:(length(dataname)-2)]) %>% ## take only a subset of the data
    rownames_to_column("COMP_ID") %>% ## make row names a column
    slice(2:n()) ## remove top line

  write.csv(data_subset, file="tmp_data_subset.csv", row.names = F)

  data_subset <- as.data.frame(read.csv("tmp_data_subset.csv", header=T))
  data_subset.na <- data_subset

  #do calculation on data
  effect.size <- (as.numeric())
  effect.size.vector <- (as.numeric())
  power.vector <- (as.numeric())
  ID.vector <- (as.numeric())
  sd.vector <- as.numeric()
  mean.vector <- as.numeric()
  mean.dff <- as.numeric()
```

```

for (row in 1:nrow(data.subset.na)) {
  temp <- data.subset.na[row,]
  temp <- log(temp, base = 2)
  ## standard calc: (mean_x-mean_y)/stdev

  # calculate effect sizes
  x.len <- as.numeric(dataname[as.numeric(length(dataname))-1])
  y.len <- as.numeric(dataname[as.numeric(length(dataname))])

  mean.x <- mean(as.numeric(temp[2:(x.len+1)]),na.rm=TRUE)
  mean.y <- mean(as.numeric(temp[(x.len+2):(x.len+y.len+1)]),na.rm=TRUE)
  sd.temp <- sd(temp[2:length(temp)],na.rm = T)
  effect.size <- (abs(mean.x-mean.y))/sd.temp

  effect.size.vector <- c(effect.size.vector,effect.size)

  sd.vector <- c(sd.vector,sd.temp)
  mean.vector <- c(mean.vector,mean.x,mean.y)
  mean.dff <- c(mean.dff,(abs(mean.x-mean.y)))
}
}

```

Mean

Means

```
print (summary((mean.vector)))
```

```
##      Min. 1st Qu.  Median    Mean 3rd Qu.    Max.   NA's
##  14.04  17.70   19.40   20.00  22.02   30.15   154
```

```
hist((mean.vector),xlab="Mean (Log2)", main = "Compound levels observed in data",25)
```

Compound levels observed in data

Differences between means

```
print (summary((mean.dff)))
```

```
##      Min. 1st Qu.  Median    Mean 3rd Qu.    Max.   NA's  
## 0.00262 0.10812 0.25350 0.37595 0.53145 3.82620     82
```

```
hist((mean.dff),xlab="Mean difference (Log2)", main = "Differences in mean observed in data",25)
```

Differences in mean observed in data

```
print (summary(exp(mean.dff)))
```

```
##   Min. 1st Qu.  Median    Mean 3rd Qu.    Max.   NA's  
##  1.003  1.114  1.289   1.698  1.701 45.888    82
```

```
hist((2^mean.dff),xlab="Fold change", main = "Fold changes observed in data",50,xaxp = c(0, 20, 10))
```

Fold changes observed in data

Standard deviation

```
print(summary((sd.vector)))
```

```
##      Min. 1st Qu.  Median    Mean 3rd Qu.    Max.   NA's  
## 0.0019 0.4445 0.6002 0.6343 0.7867 2.2388    77
```

```
hist((sd.vector),xlab="Standard deviation (Log2)", main = "Standard deviations observed in data",25)
```

Standard deviations observed in data

Representative values

```
mean <- 20.00
stdev <- 0.6343
z <- seq(1,10, by=0.5)
z.log <- log(z, base = 2)
```

The mean and stdev were calculated from all the data following log transformation.

Power cacluations

I calculated the effect size for compounds in data. Where effect size is $\text{mean}_x - \text{mean}_y / \text{standard deviation}$.

```
hist(effect.size.vector,xlab="Effect size", main = "Observed effect sizes in data",50)
```

Observed effect sizes in data

Standard deviation

For this analysis I used the quartiles to determine the standard deviation.

```
print(summary((sd.vector)))
```

```
##      Min. 1st Qu.  Median    Mean 3rd Qu.    Max.    NA's  
## 0.0019  0.4445  0.6002  0.6343  0.7867  2.2388     77
```

```
## low
```

```
pwrt <- power.t.test(n = 3,  
  delta = (mean+z.log)-mean,  
  sd = 0.4445,  
  sig.level = 0.05,  
  power = NULL,  
  type = "two.sample",  
  alternative = "two.sided")  
combined.power.data <- cbind(z,pwrt$power)
```

```
## medium
```

```
pwrt <- power.t.test(n = 3,  
  delta = (mean+z.log)-mean,  
  sd = 0.6002,  
  sig.level = 0.05,  
  power = NULL,  
  type = "two.sample",  
  alternative = "two.sided")  
combined.power.data <- cbind(combined.power.data,pwrt$power)
```

```

## high
pwrt <- power.t.test(n = 3,
  delta = (mean+z.log)-mean,
  sd = 0.7867,
  sig.level = 0.05,
  power = NULL,
  type = "two.sample",
  alternative = "two.sided")
combined.power.data <- cbind(combined.power.data,pwrt$power)

# format the data for ggplot
combined.power.data <- as.data.frame(combined.power.data)
colnames(combined.power.data) <- c("z","SD 0.44","SD 0.60","SD 0.78")
combined.power.data.melt <- melt(combined.power.data,id=c("z"))

ggplot(combined.power.data.melt,aes(x=z,y=value,group=variable,color=variable)) +
  geom_point() +
  geom_line() +
  ylab("Power") + xlab("Fold change") + ggtitle("Power predictions for representative standard deviations") +
  theme_classic()+
  theme(plot.title = element_text(hjust = 0.5)) + # center the title
  theme(legend.position = c(0.9, 0.5))

```

```

rm(combinded.power.data)
## low
pwrt <- power.t.test(n = 5,
  delta = (mean+z.log)-mean,
  sd = 0.4445,
  sig.level = 0.05,
  power = NULL,
  type = "two.sample",
  alternative = "two.sided")
combinded.power.data <- cbind(z,pwrt$power)
## medium
pwrt <- power.t.test(n = 5,
  delta = (mean+z.log)-mean,
  sd = 0.6002,
  sig.level = 0.05,
  power = NULL,
  type = "two.sample",
  alternative = "two.sided")
combinded.power.data <- cbind(combinded.power.data,pwrt$power)

## high
pwrt <- power.t.test(n = 5,
  delta = (mean+z.log)-mean,
  sd = 0.7867,
  sig.level = 0.05,
  power = NULL,
  type = "two.sample",
  alternative = "two.sided")
combinded.power.data <- cbind(combinded.power.data,pwrt$power)

# format the data for ggplot
combinded.power.data <- as.data.frame(combinded.power.data)
colnames(combinded.power.data) <- c("z","SD 0.44","SD 0.60","SD 0.78")
combinded.power.data.melt <- melt(combinded.power.data,id=c("z"))

ggplot(combinded.power.data.melt,aes(x=z,y=value,group=variable,color=variable)) +
  geom_point() +
  geom_line() +
  ylab("Power") + xlab("Fold change") + ggtitle("Power predictions for representative standard deviation") +
  theme_classic()+
  theme(plot.title = element_text(hjust = 0.5)) + # center the title
  theme(legend.position = c(0.9, 0.5))

```

Power predictions for representative standard deviations
n = 5 sig = 0.05

REVIEWERS' COMMENTS:

Reviewer #2 (Remarks to the Author):

First of all, I wish the authors could provide a revised manuscript with the revisions highlighted. It could help the reviewer to check where the revisions have been made on the manuscript, and facilitate a quick evaluation of the revised manuscript.

Based on the rebuttal letter (response to Reviewer 2), I think my major concerns of the work are NOT fully addressed by the authors. For the revised manuscript, I still have two major concerns, which I think the authors will NOT be willing to improve before the final publication:

1. The whole study is too descriptive, and mechanistic validations using genetic manipulation are largely missing from the work.

2. The technical quality of the metabolomics measurement provided by Metabolon has to be improved or disclosed (may be impossible since Metabolon may not be able to provide the details).

Finally, I still have concern with sample size, statistics and data analysis of the work.

Some specific comment to the authors' response:

Comment related to "Response to comment 2&3":

1. RSD: please read my original comment carefully! The authors should provide "Relative Standard Deviation (RSD) for each compound measured in the pooled samples", not in each individual biological sample. RSD should be calculated using the pooled QC samples as the technical replicates, and maximum 30% RSD for a compound is a common requirement for quality control purpose. This is common practice for metabolomics. I do NOT ask for RSD for BIOLOGICAL SAMPLES, which was provided by the authors in the revision. I did NOT request to discard metabolite data based on RSD values of biological samples (which is non-sense). Remove the metabolites with RSD>30% in pooled QC samples, which is a common practice.

2. Missing values: metabolites with MV>50% in all sample groups should be removed from the subsequent analysis. If missing values in one sample group are less than 50%, it is OK to keep it (I agree the NAA example is totally fine). Please check this again and confirm.

Comment related to "Response to comment 4": I still believe you should use p-value cut-off of 0.05, otherwise, how could I trust your analysis? The same concern with the sample size.

Reviewer #3 (Remarks to the Author):

The reviewers have adequately addressed my concerns.

If I may remark as a small comment: I would not use the abbreviations for the diets in the titles and figures (SD etc.). I would reconsider the changes which have been made. They make them more cryptical than necessary.

We thank the reviewers for their comments and we have addressed each reviewer's comments separately below in blue.

Reviewer #2

"First of all, I wish the authors could provide a revised manuscript with the revisions highlighted. It could help the reviewer to check where the revisions have been made on the manuscript, and facilitate a quick evaluation of the revised manuscript."

In addition to the PDF file, we had submitted a MS word version of the manuscript that tracked all the revisions made. Tracked changes are also available for v3.

"Based on the rebuttal letter (response to Reviewer 2), I think my major concerns of the work are NOT fully addressed by the authors. For the revised manuscript, I still have two major concerns, which I think the authors will NOT be willing to improve before the final publication:
1. The whole study is too descriptive, and mechanistic validations using genetic manipulation are largely missing from the work."

We disagree with the reviewer, we think that this manuscript will be a very useful and sought after resource for our community. Combining it with in depth mechanistic studies would create several additional figures and a new body of knowledge that is best represented in a new manuscript.

"2. The technical quality of the metabolomics measurement provided by Metabolon has to be improved or disclosed (may be impossible since Metabolon may not be able to provide the details). Finally, I still have concern with sample size, statistics and data analysis of the work. Some specific comment to the authors' response: Comment related to "Response to comment 2&3":
1. RSD: please read my original comment carefully! The authors should provide "Relative Standard Deviation (RSD) for each compound measured in the pooled samples", not in each individual biological sample. RSD should be calculated using the pooled QC samples as the technical replicates, and maximum 30% RSD for a compound is a common requirement for quality control purpose. This is common practice for metabolomics. I do NOT ask for RSD for BIOLOGICAL SAMPLES, which was provided by the authors in the revision. I did NOT request to discard metabolite data based on RSD values of biological samples (which is non-sense). Remove the metabolites with RSD>30% in pooled QC samples, which is a common practice.
2. Missing values: metabolites with MV>50% in all sample groups should be removed from the subsequent analysis. If missing values in one sample group are less than 50%, it is OK to keep it (I agree the NAA example is totally fine). Please check this again and confirm."

With respect to the RSDs of the QC samples, there are two classes that are present, one calculated from Metabolon internal standards (to monitor instrument performance, the CMTRX matrix) and one from endogenous biochemicals detected in the QC technical replicates. The endogenous biochemical RSD is filtered on compounds with 100% fill (present in every sample)

since a median would be misrepresented if taken across biochemicals with a high degree of sparseness. A total of eight CMTRX samples were present in this analysis (all samples contributed to the Median RSD score – these are represented in the supplied PCA (difficult to see since samples tend to stack on top of each other).

QC Sample	Measurement	Median RSD
Internal Standards	Instrument Variability	3 %
Endogenous Biochemicals	Total Process Variability	8 %

In these QC samples, biochemicals with an RSD **less than** 30% were the vast majority of findings, accounting for approximately **90% of compounds detected in all CMTRX samples**. This is considered excellent performance for a global metabolomic platform. Metabolon has a strong commitment to inclusion of appropriate QA/QC parameters in experimental runs, emphasized by their active participation in the recent mQACC working group meeting (described in PMID: 30830465) hosted by the National Cancer Institute. Metabolon CMTRX (technical replicate) samples are meant to report on overall run quality (ie, instrumentation function) rather than provide a metric for stratification of observations, and as the reviewer guessed, these are proprietary to Metabolon and are not released.

To address the reviewer’s suggestion, Jason Kinchen, the scientist at Metabolon who ran and analyzed the samples presented here and who is an author in the manuscript, posed the question to a team of statisticians at Metabolon who routinely evaluate data from the metabolomic platform. This group was somewhat perplexed by the question – a high RSD will increase the chance of a false-negative observation, however our manuscript draws conclusions from statistically-significant associations (which would make this a moot point). While acquisition-related variation does impact the ability to distinguish significant effects (which may set a higher hurdle for compounds with a higher RSD), this is also a function of study power (N per group) and effect size, which was robust in our study. Setting an arbitrary RSD cut-off would eliminate useful observations without providing a substantial benefit to overall data quality. We do not agree that this arbitrary cut-off is widely used in the metabolomics (or statistics) community. First, a compound with a RSD of 40% but that showed a 2-fold change consistently would be significant in an analysis with an appropriate group size, so excluding this biochemical would lose valuable information. Second, the analysis here focused on significant changes identified – if a 30% RSD cut-off produces too much noise to detect a difference, these biochemicals have already been excluded from the analysis since a priori they could not

achieve significance by the reviewer's logic. To make this point clear in the manuscript, we would like to add this sentence to the analytical methods section, which should address any concerns around potentially misleading observations:

"An RSD cut-off was not applied to the data, which may lead to false-negative observations when considering biochemicals that did not achieve significance."

With respect to missing data, as we stated in the previous response to the reviewer's comments, we confirm that 50% missingness threshold was applied on a per-group basis was applied in our analysis. The R code used to filter the data for missing values can be found in the "preprocessing step" of the SI document.

"Comment related to "Response to comment 4": I still believe you should use p-value cut-off of 0.05, otherwise, how could I trust your analysis? The same concern with the sample size."

Supplementary Table 1 clearly shows which compounds are within the 0.05 and 0.1 cutoffs. We have made all of our raw data available and disclosed the analysis via markup documents, so readers do not have to "believe" the data; they can use all the information and the data we provided to carry out their own analysis with their particular parameters established.

Reviewer #3

"The reviewers have adequately addressed my concerns. If I may remark as a small comment: I would not use the abbreviations for the diets in the titles and figures (SD etc.). I would reconsider the changes which have been made. They make them more cryptical than necessary."

We thank the reviewer for their comments and suggestions, they have made the manuscript stronger. Whenever space allowed we have changed the abbreviations in the figures and text to full words.